# Convergent evolution in a large cross-cultural database of musical scales

**John M. McBride**[1]*, **Sam Passmore**[2,3], **Tsvi Tlusty**[1,4]*

**1** Center for Soft and Living Matter, Institute for Basic Science, Ulsan, South Korea, **2** Faculty of Environment and Information Studies, Keio University, Fujisawa, Japan, **3** Evolution of Cultural Diversity Initiative, College of Asia and the Pacific, Australian National University, Canberra, Australia, **4** Departments of Physics and Chemistry, Ulsan National Institute of Science and Technology, Ulsan, South Korea

* jmmcbride@protonmail.com (JMM); tsvitlusty@gmail.com (TT)

## Abstract

Scales, sets of discrete pitches that form the basis of melodies, are thought to be one of the most universal hallmarks of music. But we know relatively little about cross-cultural diversity of scales or how they evolved. To remedy this, we assemble a cross-cultural database (Database of Musical Scales: DaMuSc) of scale data, collected over the past century by various ethnomusicologists. Statistical analyses of the data highlight that certain intervals (*e.g.*, the octave, fifth, second) are used frequently across cultures. Despite some diversity among scales, it is the *similarities* across societies which are most striking: step intervals are restricted to 100-400 cents; most scales are found close to equidistant 5- and 7-note scales. We discuss potential mechanisms of variation and selection in the evolution of scales, and how the assembled data may be used to examine the root causes of convergent evolution.

## Introduction

Music, like language, can be described as a generative grammar consisting of basic building blocks, and rules on how to combine them [1, 2]. In melodies, these basic units are usually specified by two quantities: frequency and duration. We generally refer to this basic pitch unit as a note, and a set of notes as a scale. Thus, as far as pitch is concerned, a scale is to a melody what an alphabet is to writing. Despite their centrality to music and apparent ubiquity, we know surprisingly little about scales. Most studies focus on scales from a limited number of musical traditions [3, 4], and the main statistical findings are that scales are non-equidistant, and have 7 or fewer notes [5, 6]. There are many anecdotal reports that certain notes are commonly used [3, 4, 7–10], but this has only been quantitatively examined in one small cross-cultural sample [11]. Thus, we lack concrete understanding of the fundamental questions of why humans use scales, how diverse they are, or how they came to be that way. We suspect that this is simply due to a lack of suitable resources. Here, we address this problem by presenting and analyzing a data set of musical scales from many societies, extant and extinct, built upon a century of ethnomusicological enterprise.

Let us begin by clarifying a few key terms and ideas. We define a *scale* simply as a sequence of unique notes (Fig 1A). *Notes* are pitch categories described by a single pitch, but as pitch is

**Data Availability Statement:** DaMuSc (Database of Musical Scales) is available at https://github.com/jomimc/DaMuSc. An archived version of DaMuSc, along with data needed to recreate figures is available on Zenodo (http://doi.org/10.

5281/zenodo.7250281). Analysis code and code for creating figures is available at https://github.com/jomimc/DaMuSc_Paper_Code.

**Funding:** This work was supported by the Institute for Basic Science, Project Code IBS-R020-D1. The funders had no role in study design, data collection and analysis, decision to publish, or preparation of the manuscript.

**Competing interests:** The authors have declared that no competing interests exist.

continuous, notes are more realistically described as regions of semi-stable pitch centered around a representative (*e.g.*, mean, median) frequency [12, 13]. However, humans process relative frequency much better than absolute frequency [14], so scales are practically specified by the *intervals* (frequency ratios) between notes. Thus, a scale is usually defined by the ratios made between all notes and the first note, called the *tonic*; we hereafter refer to these as *scale notes*. Intervals are determined by their frequency ratios, but since we perceive pitch logarithmically [15, 16], they are naturally measured in units of *cents*—obtained by a base-2 logarithm of the ratio of frequencies $f_1$ and $f_2$, cents = $1200 \times \log_2 f_1/f_2$. In what follows, *steps* will describe the intervals between adjacent notes, and *intervals* can refer to the interval between any pair of notes.

We explicitly define an important sub-class of scales, *octave scales*, which span an octave (frequency ratio 2:1; 1,200 cents) and have circular (or helical) structure such that notes repeat at every octave [17]; *e.g.*, Fig 1A shows that the first and last notes (an octave apart) of such a scale have the same name—a property is called *octave equivalence*). It is often reported that octave equivalence is one of the most universal features of music [3, 4, 7–10]. However, this has been disputed [13, 16, 18–20], and statistical study of octave prevalence is still lacking. Octaves are indeed salient in the harmonic spectra [21, 22]. Yet, experiments have indicated that octave equivalence is only a weak perceptual phenomenon [23–26], that depends heavily on musical training [27–29], and culture [16]. Nonetheless, several measurable phenomena related to harmonic tones may lead to preferential use of the octave: tonal fusion [26, 30, 31], hearing in noise [32], and memorability of complex tones [33]. For example, increased tonal fusion between octave intervals (likewise for fourths, fifths, which also undergo tonal fusion) may give rise to greater perception of synchrony when singing in parallel [34].

To address longstanding questions on scale diversity and evolution, we first document the creation of a database of scales. We then analyse the empirical scales to see which intervals are significantly frequent or infrequent, providing robust statistical evidence that the octave is prevalent among many societies. We examine the cross-cultural diversity of octave scales, finding that within-society variation can in some cases be as great as the total variation in scales—scales are surprisingly not that diverse. In particular, across all geographical regions step

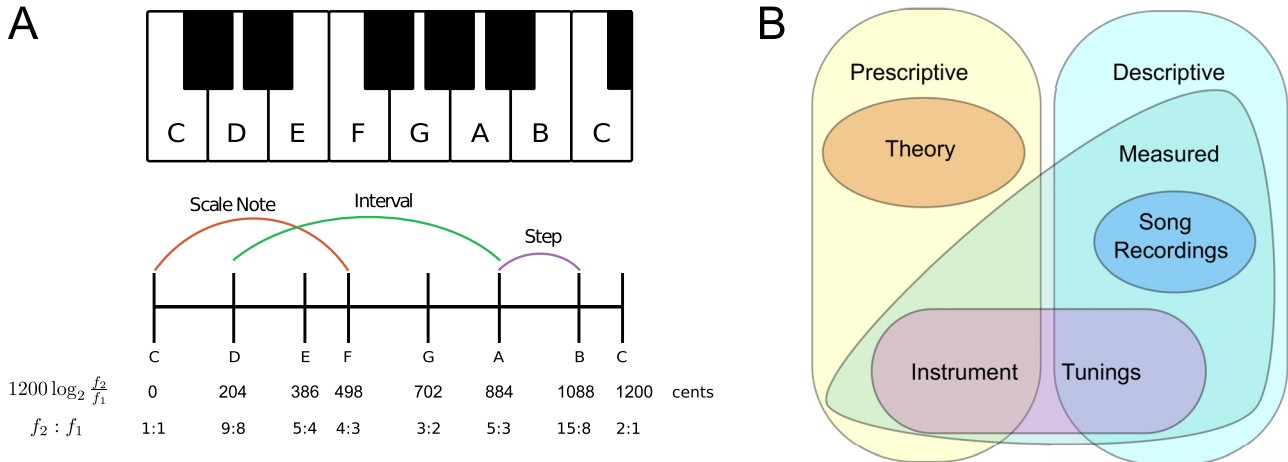

**Fig 1. Basic notions.** A: Illustration of relevant terms: Notes in a scale can be represented symbolically (*e.g.*, letters for notes), or quantitatively. As examples, we show: the Western major scale in the key of C on a piano (top); and the corresponding intervals in cents and as frequency ratios (bottom; in 5-limit just intonation tuning). We show three types of intervals: a scale *note*, a *step*, and an *interval* that is neither a note or a step. B: Venn diagram indicating how scale data can be classified.

intervals are limited to 100–400 cents, and scales are clustered around 5- and 7-note equidistant scales (scales where step sizes are similar in size). As a result of these restrictions, humans tend to use only ∼1% of possible octave scales, which strongly suggests some degree of convergent evolution due to shared biases (in addition to convergence via cultural diffusion). Finally, we discuss the potential mechanisms of change and selection of scales, discuss the challenges in understanding scales evolution, and propose credible future directions for studying this evolution.

## Materials and methods

### Database curation

A total of 60 books, journals, and other ethnomusicological sources were found to have relevant data on scales (S1 Table in S1 File) [13, 35–93]. We note one previous attempt to create a database of scales from the ethnomusicological literature [94], however there are no details on its construction, and it does not link scales directly to original sources. Our database improves on these issues through a stringent methodology (described below), sources for all scales and is presented as a digitized, open-access, resource which will continue to grow as new data is made available.

We can define scales (Fig 1B) either prescriptively ("these are the notes you can use in a melody") or descriptively ("these are the notes that were used in the melody"). *Theory* scales consist of intervals with idealized, exact frequency ratios—they lack the random fluctuations that exist in the real world. These are mainly found in a limited set of cultures that exist along the old Silk Road route, and they are not necessarily played according to these theoretical ideals [93, 95]. Absence of theory scales does not mean that other societies lack musical theory, which can be found either explicitly through language, or implicitly through an understanding of what is 'correct' and 'incorrect' in a particular style [2, 96–100]. Theory scales are by definition prescriptive scales, although descriptive scales can be found which closely match these. *Measured* scales are obtained where measurements have been made of the notes on an instrument, or a recording of a song has been analysed with computational tools to extract a scale. Instrument tunings are by default prescriptive, but can be descriptive if all notes are used in a melody. There is some error in these measurements—taken with tools that include tuning forks, the Stroboconn, and modern computational approaches—but it is always within 10 cents. Measured scales taken from song recordings are exclusively descriptive, and they make up the smallest part of the database. This is because it is still quite a challenge to reliably infer scales from a recording of a performance using algorithms [76], and thus it requires extensive manual labor. Still, we believe that the future of studies on musical scales lies in tackling these challenges, due to the prospect of extracting descriptive scales from archives of ethnographic recordings [101–104]. However, it is clear that the appropriate computational tools to perform such large-scale analyses are still lacking [105–107].

Another statistical regularity in musical pitch is *tonality*—*i.e.*, the distribution of notes, how they are played (duration, ornamentation), and their transition probabilities [109, 110]. Of the various aspects of tonality, tonal hierarchies (probability distribution of notes) [111] have been best documented across cultures [73, 112–115]. Tonality (or modality) is in some cases an integral part of a scale that determines how the notes should be played [116], to the extent that musicians can reliably tell apart two ragas from the duration of a single note [117]. We emphasise that our definition of scale does not imply any specific tonal hierarchy, while acknowledging that similar definitions (*e.g.*, maqam, raga) are often used with the expectation that they include tonality [100]. In the database, we account for tonality in a limited way by noting the order of notes in the scale, starting from the first note (tonic). However, this information is

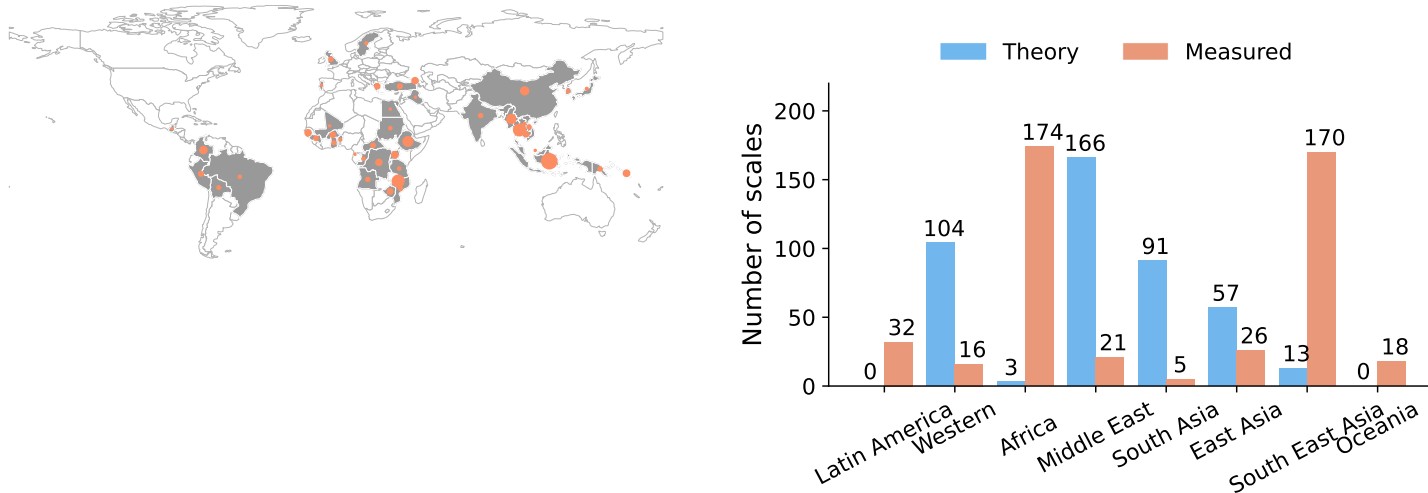

**Fig 2. Breakdown of scales in the database according to: Geographical area; theoretical scales or measured scales.** The map shows the geographic origin of the measured scales; theory scales are not included since they are not always associated a single country; marker size indicates sample size. Made with Natural Earth [108].

often unavailable, and we lack the finer details of note distribution or transition probabilities. Future cross-cultural analyses of recordings ought to document these important details [116, 118].

To enable a broad range of analyses we collected information pertaining to, where applicable, (i) the society (country, language or ethnic group, musical tradition), (ii) geography (country, geographic region) (Fig 2), (iii) instrument type, (iv) tonic note, and (v) whether the scale is measured using harmonic or melodic intervals. We additionally linked societies, where appropriate, to identifiers from other ethnographic and linguistic databases, such as D-PLACE and Glottolog [119, 120]. D-PLACE contains information on social structure, economic, and environmental information, while Glottolog can be used to determine the linguistic distance between two groups—a common proxy for cultural similarity [121]. Some musical traditions that span multiple countries (such as Western classical music) are also taken as a unit of society. Identification of the society or geographic origins of a scale was required for inclusion of scale data. Other details were found at varying frequency; *e.g.*, tonic was only identified in 121 out of 434 measured scales.

### Inferring octave scales

The database exists in two forms: the raw data (434 theory scales, and 462 measured scales), and a set of octave scales that is generated procedurally from the raw data according to five choices. For a complete workflow from source to database, including examples, and the choices we made to generate octave scales, see the supplementary materials(S1 Appendix, S1-S3 Figs and S1, S2 Tales in S1 File).

In total, we infer 896 octave scales (434 theory octave-scales, 384 octave-scales from instrument tunings, and 78 octave-scales from song recordings), from 73 societies. The theory scales span 6 regions, while the measured scales span 8 regions, and 46 countries (Fig 2). When inferring octave scales from measured scales, a principle assumption is that they add up to an octave. Since the validity of the assumption is not clear, we first study the statistics of the measured scales before studying octave scales.

## Results

One of our goals is to estimate whether certain notes and intervals appear more or less frequently than expected by chance. This is hard to quantify, since we lack a universally-correct method of calculating the probability of observing an interval. Instead, we propose three independent statistical models that reflect different ways of constructing scales:

### Model: Lognorm

This model assumes that scale notes $S$ are chosen independently from a lognormal distribution, $P(S) = \ln N(\mu, \sigma^2)$. The rationale behind this choice is that small intervals should be uncommon ($P(S) \rightarrow 0$ as $S \rightarrow 0$) due to limits in pitch perception, and large intervals should be uncommon ($P(S) \rightarrow 0$ as $S \rightarrow \infty$) due to physical constraints (human anatomy; instrument size). We expect that this model is inappropriate for within-scale analyses, since notes within a scale are unlikely to be chosen independently of each other, but rather according to specific choices made by a musician. However, if the choices made by many independent musicians are sufficiently diverse, then the lognormal distribution may be an appropriate description of the between-scales note distribution.

### Model: Shuffle

This model assumes that specific step sizes are important, but their arrangement is not. Normally, both step size and order determine scale notes. But now consider a musician who cares only about the step sizes, not their order in which they are arranged. This can be approximated by sampling from the original data and reshuffling the order of step sizes.

### Model: Resample

This model assumes that step sizes are chosen independently from a distribution, and arranged randomly into scales. This is equivalent to a musician who is indifferent to both the step size, and the order. A reasonable choice of a distribution in this case is the posterior distribution of all step sizes.

In any alternatively-sampled set of scales, the number of scales and the number of notes in these scales matches the original. In the following sections, we will use these three models to, first, compare intervals between scales, and second compare intervals within scales, searching for evidence of statistically significant counts of intervals.

### Statistically significant intervals between scales

To search for statistically significant (due to an abundance or dearth of) intervals, we first plot the distribution of scale notes (Fig 3A, Original) and step sizes (Fig 3B, Original) for a subsample including *only* measured scales, without imposing octave equivalence, and controlling for society (214 scales sampled from a total of 434 scales, with no more than 5 per society, resampled 1,000 times to achieve convergence; S4 Fig in S1 File). We compare this empirical distribution to the maximum-likelihood lognormal distribution (Fig 3A, Lognorm), and the corresponding step size distribution (Fig 3B, Lognorm). As expected, we find that scales drawn from the lognormal distribution have step sizes that do not resemble real scales—since notes within a scale are not independent. Instead, notes within scales are more evenly distributed, such that step sizes are peaked at $\sim 200$ cents, and rarely smaller than 100 cents (Fig 3B); this is found in every geographical region that we investigated (S5 Fig in S1 File). However, scale notes from *different* scales are independent of each other, resulting in a distribution that is approximately lognormal when many scales are sampled (Fig 3A).

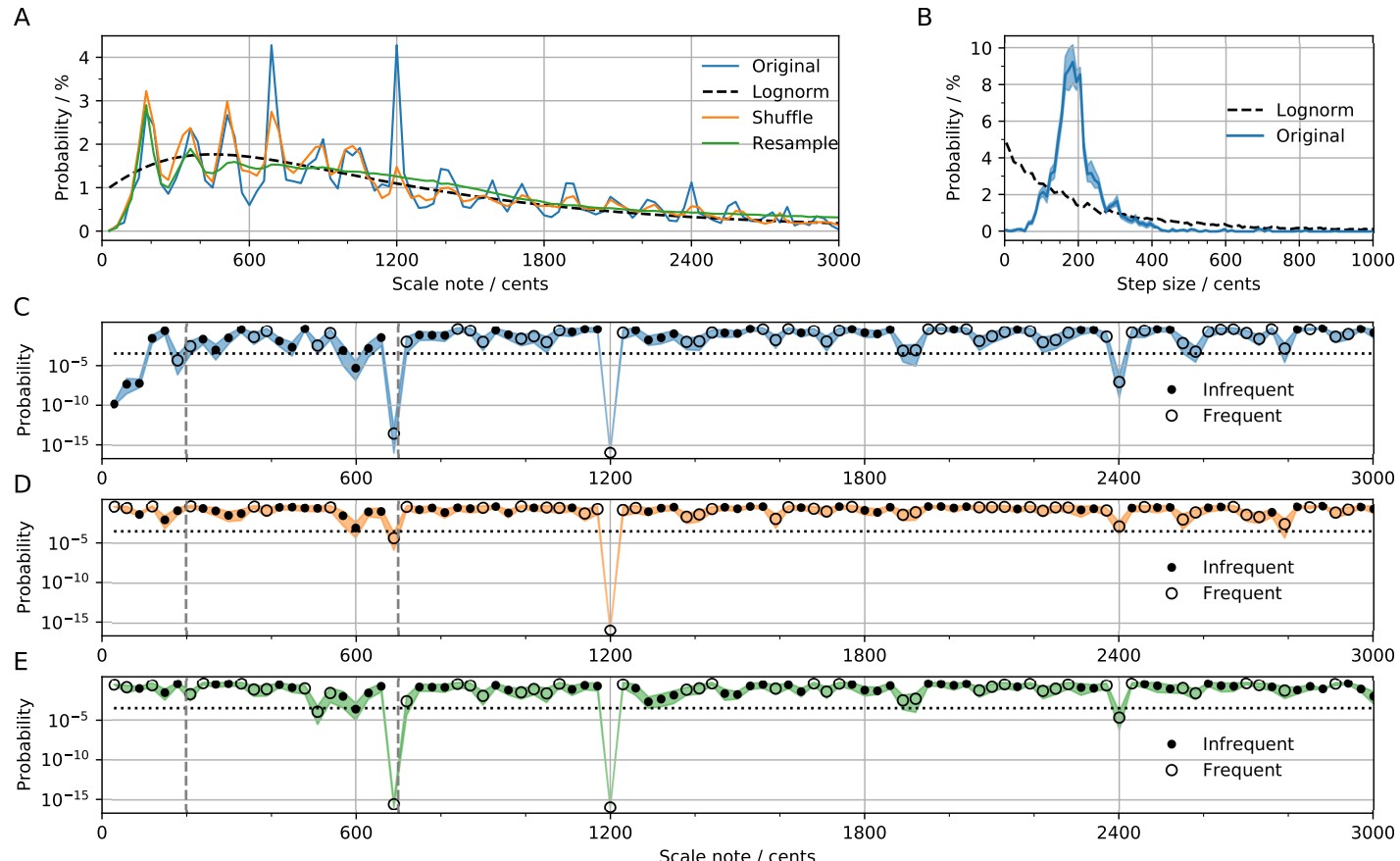

**Fig 3. Between-scales distributions of step sizes and scale notes.** A: Distribution of notes in measured scales (blue; histograms are shown as lines; bin size = 30 cents). Maximum likelihood lognormal distribution fitted to notes distribution (black). Distribution of notes obtained via alternative sampling: shuffling the step sizes within scales (orange); resampling step sizes from the full distribution of step sizes (green). The x-axis is truncated at 3,000 cents for clarity. B: Distribution of step sizes (blue). Distribution of step sizes in a set of scales with notes generated independently from a lognormal distribution. C-E: Probability that the counts observed in the original data were generated by the lognormal distribution (C), the shuffled scales distribution (D), or the resampled scales distribution (E). Empty circles indicate that the interval is found more than chance, while filled circles indicate the opposite. Dotted line indicates a *p* value of 0.05 after applying a Bonferroni correction. Dashed lines indicate the values of 200 and 700 cents. Shaded region shows bootstrapped 95% confidence intervals.

The utility of fitting a lognormal distribution to the data is that it can serve as a null-hypothesis baseline for estimating the probability that a scale note appears more or less than chance. To this end, we integrate the lognormal distribution over the range of each histogram bin $i$ in Fig 3A to get the probability of observing scale notes, $p_i$. We then calculate the binomial probability, $q_i = \binom{n}{k_i} p_i^{k_i} (1 - p_i)^{n-k_i}$, where $k_i$ is the number of observations in bin $i$, and $n$ is the total number of observations. We report either: the probability that $k_i$ or higher is observed if $k_i/n > p_i$, $\sum_{j=i}^{n} q_j$; or else, the probability that $k_i$ or less is observed, $\sum_{j=0}^{i} q_j$, if $k_i/n \leq p_i$. Low probability implies significant deviation from the null lognormal hypothesis. We see that only a few intervals deviate significantly from the lognormal distribution (Fig 3C): 1,200, 700, 200 and 2,400 cents are found more frequently, while 600 cents is found less frequently than chance. This is strong evidence that the octave (and the fifth) are important intervals in many societies.

To corroborate these findings, we repeat the significance test with different assumptions on how scales are generated. We first repeatedly shuffle the step sizes in each scale to generate new scales, to examine whether the statistically significant intervals would arise if step sizes

were ordered randomly (Fig 3D). Similarly, by resampling with new step sizes, we can test whether the significant intervals could plausibly have been produced by arranging randomly selected step sizes from the distribution in Fig 3B (Fig 3E). These analyses demonstrate that the peak at 200 cents (Fig 3C) is due to it being the most common step size. The values of 600 and 2,400 cents are found to be less significant than in Fig 3C, but are still much more significant than most intervals. The fact that the peak at 700 cents is significant in Fig 3D but not in Fig 3E is likely due to the presence of equidistant scales (S6 Fig in S1 File), where the order of the intervals is mostly irrelevant. By this logic, one might expect the peak at 1,200 cents to disappear in Fig 3D, but it survives because a large fraction of equidistant scales do not extend beyond the octave (S6 Fig in S1 File). Remarkably, the octave is extremely significant according to all three tests.

## Statistically significant intervals within scales

The previous method was used to assess whether intervals are found more than expected by chance across a collection of scales. Since grouping together scales from different societies might be problematic, we require a test that can discriminate unusually frequent / rare intervals within individual scales. To this end, we compare the original scales with many alternative scales that are created by shuffling the original scales' step sizes. We use the full range of intervals that can be generated with an instrument, not just the scale notes. For an instrument with $N$ notes, this results in $N \times (N-1)/2$ intervals. By sampling many more intervals, this test is powerful enough to detect, in some cases, significant signals within individual scales.

To test for statistical significance of an interval $I$, we find all intervals in a scale that fall within $w = 100$ cents of $I$, and calculate their distance from $I$. We repeat this process for 50 shuffled versions of the scale, collecting all intervals within $w$ of $I$ in a second group. We then use a Mann-Whitney U test to examine whether either set of intervals, original or shuffled, is significantly closer (*i.e.*, $p < 0.05$) to the octave than the other (repeating 100 times to get a converged average). We demonstrate this procedure for $I = 1,200$ cents (Fig 4A): in most scales the original intervals are closer to an octave than the shuffled scales, although only a fraction of results are significant. Reasons for non-significant results include small sample sizes and the tendency of equidistant scales to produce similar intervals when shuffled.

We extend this analysis to all intervals over the range $200 \leq I \leq 2600$ cents, showing the fraction of significant results indicating an interval is found more or less frequently than chance (Fig 4B). Without a doubt, the most significant interval is the octave (35% significantly close), and the intervals that are most significantly avoided in scales are the regions flanking the octave. In further agreement with the between-scales analysis, the next most significant regions are those around 500, 600 cents, and 700 cents.

To put these results in perspective, we repeat the test on sets of scales generated by resampling step sizes from Fig 3B (Fig 4B, Null). The null distribution converges to 5% as expected, which clearly demonstrates that the high fractions of significant results in the original scales are not artefacts due to testing multiple hypotheses. Additionally, we show that the results do not depend on our choice of $w$ (S7 Fig in S1 File). The consistency of these test results reinforces the conclusion that these significant intervals are not chosen randomly.

## Effect of tuning variability on the search for universal intervals

Observation of significant intervals likely reflects the intentions of the musician to tune to these intervals. However, non-significant results do not necessarily indicate lack of such intentions, as imprecision in tuning (and tuning measurements) may result in false negatives. To understand the impact of imprecision, we generate a test set of scales (by sampling step sizes

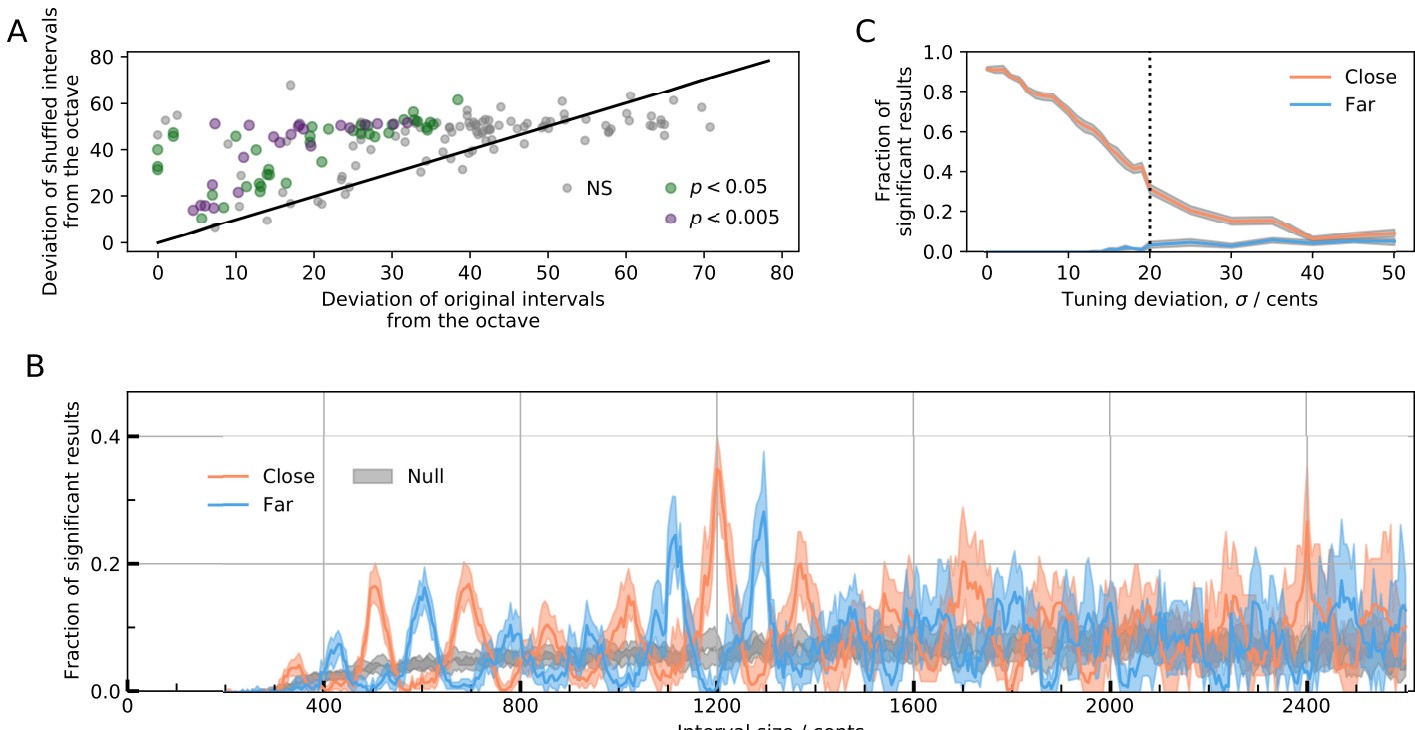

**Fig 4. Within-scales tests of interval significance.** A: Mean deviation of original intervals from the octave compared to shuffled intervals for 162 measured scales; colour indicates statistical significance; black line indicates $x = y$. B: Fraction of tested scales in which intervals are found to be significantly closer or further from a particular size than expected by chance. Null distribution (grey region) is obtained by repeating the analysis with resampled sets of scales. C: Fraction of significant results indicating that the intervals are closer or further to the octave than chance, against intonation error $\sigma$, in a test set which maximises the number of octave intervals. Dotted line indicates the point at which 35% of results are significant. Shaded region shows bootstrapped 95% confidence intervals.

from Fig 3B) and fix all scale notes that are $\geq$1,200 cents to be exactly an octave higher than one of the notes that are $\leq$1,200 cents. We then add to the test sets normally-distributed noise, $N(\mu = 0, \sigma^2)$. Even without noise ($\sigma = 0$), this test can find significant results only 90% of the time (Fig 4C), which demonstrates the difficulty in inferring intentionality simply due to low sample sizes.

To estimate reasonable bounds on the noise $\sigma$, one may first consider that the measurement error varies from about 1 cents for computational methods (*e.g.*, Stroboconn [122]), to 5 cents for tuning forks. Also, instruments vary in the stability of their notes: *e.g.*, humans typically sing with a standard deviation of at least about 10–20 cents [12, 13, 114, 123–127]. However, the main source of error is perception: humans make mistakes in discriminating intervals below $\sim$ 100 cents (although we lack replications with non-Western participants) [128–131]. Thus, we can expect a reasonable upper bound to the proportion of significant results that can be detected of about 30 to 70%. Thus, when we find the octave to occur significantly more than chance about 35% of the time (Fig 4B), this is a rate that is within the range of the hypothetical maximum.

Ultimately, it is hard to say how exactly scales are chosen. Were some important intervals fixed first (*e.g.*, the octave), and then the rest chosen to fill the gaps? Or were step sizes of a certain size (*e.g.*, big and small size categories) chosen, and then arranged in some preferred order [132]? What we have conclusively shown is that independently of how scales are created, they show a significant, consistent bias towards including some notes, and avoiding others.

## Qualitative evidence for preferential use of octaves

A more direct route to understanding the intentions of musicians is through detailed ethnography. We therefore examined each source for qualitative evidence indicating preferential use (or absence) of the octave (SI Dataset 1). We found qualitative evidence in support of octave use in 26 out of 60 sources: use of the octave to tune instruments (8 sources); performing melodies in parallel octaves (11 sources); using the same name for notes an octave apart (10 sources), which is also strong evidence for octave equivalence. We also find quantitative evidence (statistically significant octaves for at least one scale; Fig 4A) in 26 sources; in total, 40 sources contain either quantitative or qualitative evidence. For 6 of the remaining 20 sources, there is evidence from secondary sources for preferential use of the octave in those cultures; another 3 sources report on archaeological findings, which are difficult to investigate further [48, 57, 63].

Two sources (Georgian polyphonic singing) provided qualitative evidence that fifths are perceptually important, instead of octaves [13, 89]; however, we found statistically significant octaves in one of these sources, so the evidence here is mixed [13]. A third source (Colombian marimba) provided quantitative evidence that octaves were found significantly less than chance [85]. Overall, we find some evidence (either primary or secondary) in support of octaves being significant in 46 out of 60 sources, and evidence to the contrary in 3 sources. These findings strongly support the view that octaves are widespread, but not absolutely universal [19, 20].

## Statistics of octave scales

After verifying that preferential use of octaves is indeed widespread, we proceed to studying octave scales; we infer octave scales by making some assumptions (*e.g.*, octave equivalence; see section "Inferring octave scales"). While in the previous section we exclusively examined measured scales, here we look at a mix of theory and measured scales. Data is shown for four samples of scales: (i) all theory scales, (ii) all measured scales, (iii) sub-sample controlling for society ('SocID', no more than 5 scales per society), and (iv) a sub-sample controlling for geographical region ('Region', no more than 10 scales per region).

Most scales are found to have 7 or fewer notes (Fig 5A), in agreement with previous work [6]. 6-note scales are remarkably rare across all sampling schemes. The predominance of step sizes of ∼200 cents (Fig 3B) is also seen in octave scales (Fig 5B). Consistent with existing literature [2], most steps are between 100–400 cents, and this applies in all geographic regions studied (S5 Fig in S1 File). The most common notes (Fig 5C), in all sub-samples, are exact matches for the significant intervals in Fig 4B (500, 700 cents). The fact that the statistics of octave scales is consistent with the statistics of measured scales (and with previous work), to an extent, validates our methodology for extracting octave scales (SI section "Inferring octave scales").

Theory scales show sharp peaks at intervals close to 12-TET intervals. In contrast, the distributions of steps and notes in measured scales are much more diffuse. This difference is expected, since theory scales consist of mathematically-exact, ideal intervals, while measured scales include natural sources prone to error and variation. There is, however, correspondence between theory and measured scales in the rarely used notes (270, 450, 550, 650, 930 cents), which is even clearer when controlling for the number of notes (S8 Fig in S1 File). Regardless of the number of notes in a scale, we find salient peaks at 200, 500 and 700 cents (S8 Fig in S1 File). Overall, despite some differences between theory and measured scales, both sets are in agreement over the most distinct features: (1) step sizes are usually ∼200 cents, and between

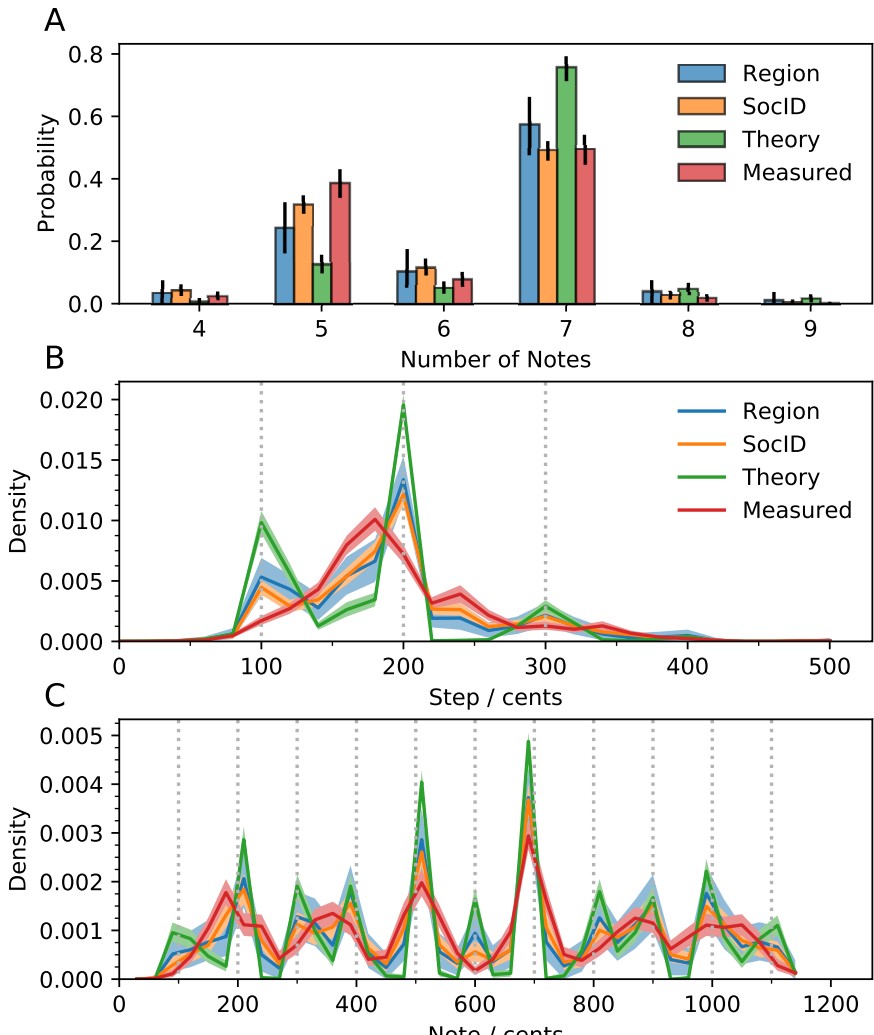

**Fig 5. Statistics of octave scales.** A: Distributions of the number of notes in a scale for different samples of the scale database: sample balanced by region, sample balanced by culture, theory scales, and measured scales. B-C: Histograms of step sizes (B) and notes in scales (C) for different samples of the scale database: samples balanced by region, samples balanced by society. Whiskers (A) and shading (B,C) indicate bootstrapped 95% confidence intervals. Histogram bins are 20 (B) and 30 (C) cents.

100–400 cents, and (2) the most salient notes are at 500 and 700 cents, while the regions around these notes are avoided in scales.

## Variation across societies is comparable to variation within some societies

To quantify variation in scales across societies, we use t-distribution stochastic neighbour embedding (tSNE) to map the scales onto a reduced two-dimensional representation [133]. This method can compare only scales with the same number of notes, so we show results separately for 7-note scales (Fig 6A), and 5-note scales (S9 Fig in S1 File). We group scales into clusters (using the DBSCAN algorithm (eps = 2, minimum samples = 5) [134], and report note distributions and region distributions for the largest four clusters. The tSNE embedding is useful for visualizing diversity among high-dimensional objects, but note that tSNE dimensions

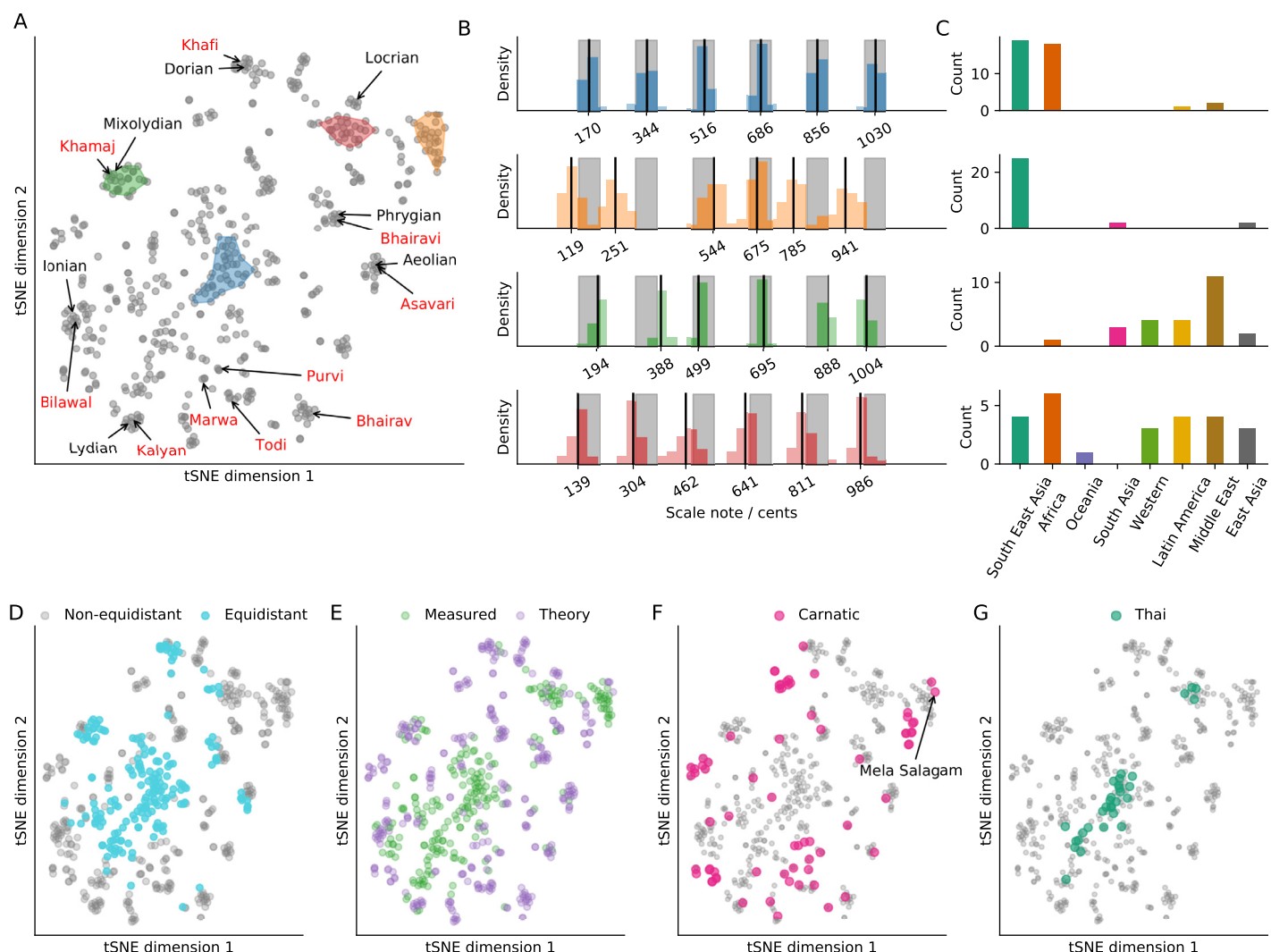

**Fig 6. Cross-cultural diversity of scales.** A: 2-dimensional embedding of 556 heptatonic scales, labelled with 7 Western diatonic modes (black), 10 North Indian thaat (red), and the four largest clusters (shaded areas). B: Note distributions for each cluster. Black lines are shown for means of each note, and grey shading indicates equidistant scale notes ±30 cents. C: Geographic distribution of each cluster. D-G: Embeddings are labelled with: equidistant scales (D; where notes are on average within 30 cents of the equidistant values), theory vs measured scales (E), Carnatic scales (F), Thai scales (G).

are arbitrary. The main information in this plot is the distances between points: distances in the 2d-embedding are non-linearly related to the real distances, such that 2d distance from one scale to all other scales will have a high rank-correlation with the real distances.

The simplest and most salient feature is that scales tend to be almost *equidistant* (Fig 6D). For example, in the largest cluster (Fig 6A and 6B; blue), notes are all within 30 cents of the corresponding notes in the equiheptitonic scale. Although most examples (29 / 40) of this cluster (Fig 6D) come from three countries (Thailand, Guinea, Malawi), in total they are found in 13 countries (*e.g.*, Colombia, Georgia, Zimbabwe, Indonesia). Surrounding this cluster (Fig 6A), we can find examples of theory scales: Western diatonic modes, and North Indian thaat (*e.g.*, green cluster). In general, we see that most equidistant scales are measured scales, but there is also a lot of overlap between theory and measured scales (Fig 6E). Furthermore, when clustering societies by scale similarity, we find two main clusters—one dominated by societies

with theory scales, and one dominated by societies with equidistant scales (S10, S11 Figs in S1 File). Examples of scales that are furthest from equiheptatonic include the Gamelan pelog scale (Fig 6A and 6B; yellow cluster), and the Carnatic mela salagam (Fig 6F), which both have many small step sizes in a row.

Surprisingly, there seems to be little variation overall, such that multiple societies that use theory scales exhibit levels of within-society variability comparable to the total variability (Fig 6F). In general, we find that geographical regions tend to contain overlapping subsets of scales (S12A Fig in S1 File), and distances between scales within-regions are of similar magnitudes to distances between scales between-regions (S12B Fig in S1 File). Even Thai scales, which are often cited as exclusively being in equiheptatonic tuning [47, 62, 69, 70]—although this is disputed [68]—exhibit substantial within-society variability (Fig 6F). Overall, there appears to be less variation in 7-note scales than might have been expected, and a similar analysis of 5-note scales reveals similar results (S8 Fig in S1 File), which suggests that they share the same organizing principles.

## Statistical analysis shows that scales tend to be equidistant

To put the diversity of scales in a broader context, one may consider a hypothetical universe of possible scales by enumerating them on a grid (*grid scales*). To this end, we take 20 cents as a basic grid resolution, and enumerate all possible unique scales with step sizes within 60–320 cents; this limitation on step size already reduces the number of possible scales (at 20 cents resolution) by 98%. To conveniently compare grid and real scales, we examine their 2-dimensional embedding. Strikingly, those grid scales that correspond to real scales (*i.e.*, within ±10 cents of a real scale) are clustered around the equiheptatonic scale (Fig 7A, cyan). For comparison, Thai scales are found to be as far as 43 cents from equiheptatonic. We find that 70% of real scales are within this boundary, compared to only 10% of grid scales (Fig 7B). Alternative definitions of equidistance based on step sizes rather than scale notes also support the finding that scales tend to be close to equidistant (S13 Fig in S1 File).

To gain further insight into the striking (near-) equidistance of scales, we look at the note distributions for notes 2–7 (Fig 7D) for real scales and grid scales. The grid scales show the broadest note distributions for the two notes (4 and 5) furthest from the fixed ends (tonic, octave), resulting in higher entropy (Fig 7E). This is expected, since summing two random distributions should result in a higher variance; *e.g.*, note 4 is the sum of three steps drawn from Fig 7C. In contrast, distributions of notes 4 and 5 in real scales are most predictable (lowest entropy), once again illustrating the significance of intervals of size 500 and 700 cents.

To examine the possibility that the difference between real scales and grid scales is mainly due to differences in their step size distributions (Fig 7C), we analyse two other sets of note distributions. We first look at the note distributions that ought to be farthest from equidistant scales: we rearrange the steps in each real scale so that they are ordered low-to-high, and consider scales starting from every position. This results in note distributions with entropy similar to grid scales (Fig 7E, Sorted). We then look at the note distributions of alternative versions of real scales obtained by shuffling the step sizes in scales. Again, we find that the entropy is highest at notes 4 and 5 (Fig 7E, Shuffled), in contrast to real scales. The main difference between the real scales and grid scales appears to be that step sizes in real scales are well-mixed—small steps are found adjacent to large steps rather than small steps (and vice versa). We found similar results for 5-note scales (S14 Fig in S1 File), which again suggests that these findings may generalize to other scales.

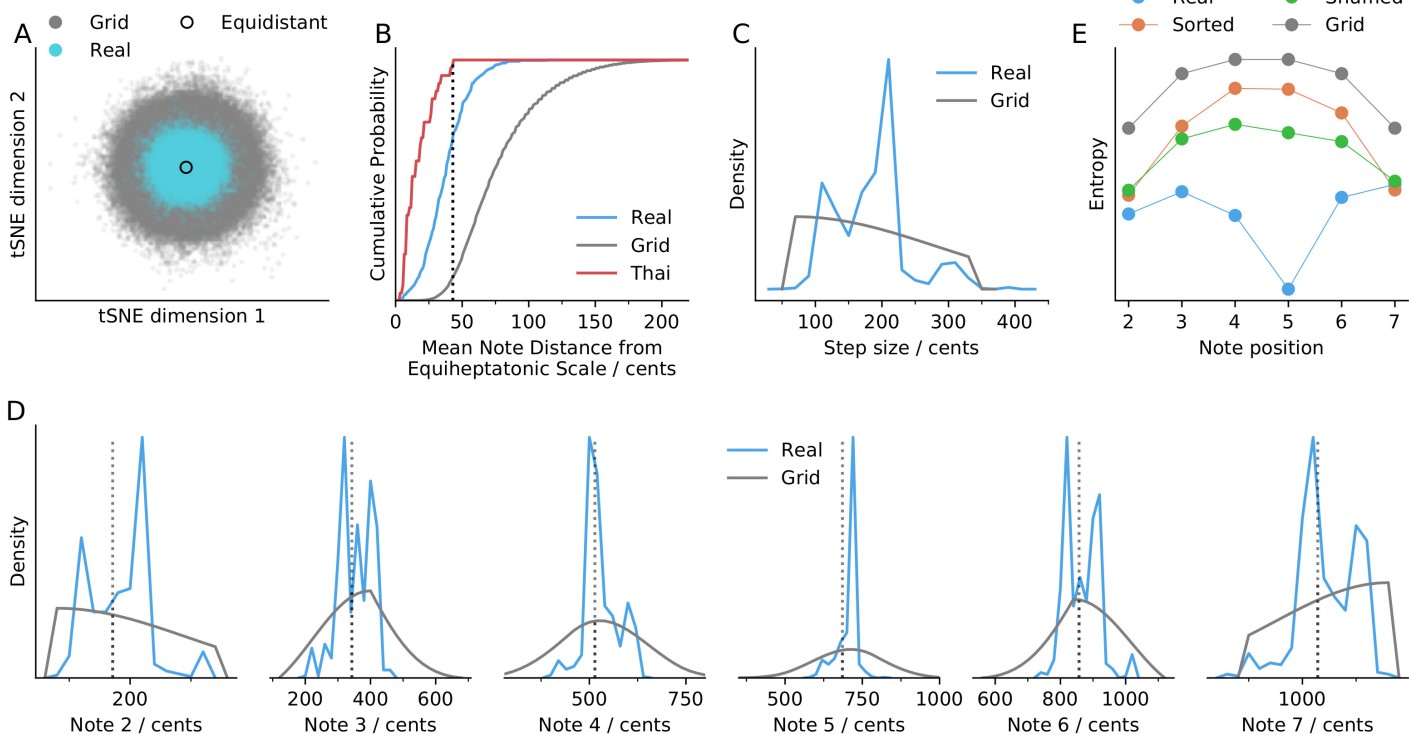

**Fig 7. Comparison of real scales with all possible 7-note scales enumerated on a grid (20 cents resolution; with step sizes limited to 60–320 cents).** A: Two-dimensional embedding of grid scales. Grid scales that correspond to real scales (notes are on average within 10 cents of a real scale) are highlighted cyan; a black circle shows the equidistant scale. B: Mean note distance from 7-note scales and the equiheptatonic scale, for Thai scales, real scales, and all grid scales. C: Step size histograms (bin size = 20 cents) in real scales and grid scales. D: Scale note histograms (bin size = 20 cents) for notes 2–7 (no tonic and octave) for real scales and grid scles. E: Entropy of note distributions for: real scales; real scales, but rearranged with their steps arranged in order of size (Sorted); real scales, but rearranged with their steps in random order (Shuffled); grid scales.

## Discussion

### How diverse are scales?

Despite some differences, scales across cultures are *remarkably similar*. For example, the set of Carnatic scales alone is almost as varied as the total set of scales (Fig 6F). In particular, most scales observed in cultures—when compared to the universe of possible scales—are close to 5- and 7-note equidistant scales (Fig 7, S12 Fig in S1 File). Many authors have reported that equidistant scales are rare [2, 6, 10, 135–137]. Yet, we find that they are more prevalent than expected by chance. This discrepancy may be due to the lack of a robust definition for equidistant scales—for example, the only previous statistical study of prevalence of equidistance does not explicitly account for natural variation in intonation [6]. It is not clear how much deviation from perfect equidistance renders a scale perceptually non-equidistant, or how variability in pitch affects perception of equidistance. As an illustration of the difficulties, consider the following example from Georgian singing: the step sizes in this scale are close to equidistant (163–202 cents) when viewing the melodic pitch histogram. But the intervals between notes in the melody are much less exact (90–240 cents) [13]; despite being statistically equidistant, it is not clear whether this scale would be perceived as equidistant. Perception of equidistance may also vary with culture and training, in which case it may be better to avoid binary measures of equidistance, and instead to construct a perceptually-relevant, continuous measure.

The most salient difference between societies is whether their data consisted of theory scales or measured scales (S9, S10 Figs in S1 File), considering that theory scales appear to be less equidistant than measured scales. One hypothetical explanation for the predominance of non-equidistant scales in societies using theory scales: the process of creating new scales by combining simple-integer-ratio intervals will inevitably result in more non-equidistant scales because only one combination of step sizes can result in an equidistant scale. We wonder whether this societal difference would persist if we were to only compare measured scales from societies, since theory scales will certainly exhibit intonation variability when performed. In the literature, Western scholars often discuss variation in intonation in societies that lack mathematical musical theory as being an intentional form of expression [64, 68, 138–141], whereas studies of classical musics typically investigate to which theoretical tuning system the musicians conform [13, 123, 126, 127]. Ultimately, it is difficult to intuit the differences between societies, since there are only a few cross-cultural studies of interval discrimination [142, 143]. Here, we performed a preliminary study on intonation variability (S15, S16 Figs in S1 File), finding comparable levels of variation (with the possible exception of the pelog scale) in Gamelan orchestras, Thai xylophones, Turkish ney [91], Georgian singing [89], and a Belgian carillon [144]. But ultimately, to understand the differences between societies that use theory scales and those that do not, we will need cross-cultural perceptual experiments, and direct measurements of scales from recordings.

How did far-away societies come to use such similar scales? – Using a cultural-evolutionary framework [145], we can describe this process as a combination of cultural diffusion, and convergent evolution due to common factors biasing the use of scales across cultures. Diffusion can certainly account for some similarities, as some societies have documented shared history: *e.g.*, societies with theory scales in the Middle East, or East Asia; court music in Thailand, Laos and Cambodia [100]. Some have suggested that cultural diffusion can be inferred based on two cultures using similar equiheptatonic scales [80, 85], but evidence seems circumstantial given how widespread equiheptatonic scales are. Instruments, on the other hand, are a richer source of information than scales, so organology studies are a much more direct way of inferring cultural diffusion [146–149]. While there is undoubtedly some transmission of information across cultures, we reiterate that: within-region scale diversity is comparable to between-region diversity (S12B Fig in S1 File), step intervals consistently show approximate limits of 100–400 cents across regions (S12B Fig in S1 File), and scales are (against chance odds) overwhelmingly close to equidistance in all regions (S12 Fig in S1 File). Taken together, these facts point towards some non-negligible degree of convergent evolution due to shared biases.

## How do scales evolve over time?

Scales can change, or persist, in a variety of ways. On a short time-scale, vocal (or other non-fixed-pitch instrument) scales are inherently stochastic, due to a lack of precision in motor control [150]. On a longer time-scale, vocal scales reside in memory, which introduces another mechanism of change. Unlike vocal scales, instrument tunings physically persist through time, but are still affected by multiple factors: environment (temperature, humidity), material (wood, metal, animal organs), and physical force [49]. To illustrate this point, we checked examples of repeated tunings of the same instruments across a specific time frame: Gamelan orchestra (metal idiophone), with standard deviation of notes, $\sigma = 8$ cents (slendro) and $\sigma = 13$ cents (pelog) cents over about 25 years [46]; Angolan likembe (plucked metal idiophone), where $\sigma = 18$ cents over a few weeks [49]; Gambian kora (chordophone), with $\sigma = 27$ cents over one week [52]. Technology offers more robust ways of keeping scales stable over time: monochords (ancient Greece) and pitch pipes (ancient China) enable reliable tuning by

fourths and fifths [36]. Likewise, the perceptual phenomenon of tonal fusion may have enabled stable scales by providing knowledge of perceptual anchors (octaves and fifths) that can be reliably passed through generations. Recently, it seems to us that stability in scales has been reinforced though global tuning standards and inventions such as fixed pitch instruments and electric tuners. Thus, we can see that scales change in many ways, but it is possible for the rate of change to decrease through the use of technology and musical theory.

We can imagine some possible mechanisms of scale evolution, drawing analogies with biological evolution: We can think of a single scale as a gene, and a set of scales used by a population of people as a genome. Changes to the scale notes is akin to mutating a single nucleotide. Adding or removing notes from a scale is like insertions/deletions of nucleotides. Scales can be copied through interactions between populations; genes can be shared through horizontal transmission. The Western diatonic modes can be constructed using the same step intervals, but changing the tonic position; the same process occurs in protein sequences (circular permutants), likely through gene duplication and homologous recombination.

The analogy starts to break when you consider that scales can be invented, which is more akin to designing genomes. There are recent examples of invention of microtonal tunings in Western music [36, 64, 151], and many theory scales bear the hallmarks of design: Greek modes are all circular permutants, based on simple integer ratios [152]; the Carnatic melakarta result from combinatorial enumeration of a set of intervals, constrained by a set of rules [153]. Similarly, other cultures that have a long history of mathematical scholarship [154] use overlapping sets of scales based on theoretical divisions of the octave [95, 155–159]. Typically, though, the theory scales that still survive are similar to the meaured scales, so it is likely that designed scales are constrained by similar selection criteria that applies to non-designed scales.

We propose that a detailed mechanistic model of scale evolution is possible: The mechanisms of spontaneous change need to be studied at the resolution of a single mutation step (through, *e.g.*, transmission chain experiments) [160]. The relative importance of the different mechanisms could be investigated with sufficient long-term data (*e.g.*, same individual/group/culture, over a period of months/years). Such a model could then be used for agent-based modelling to study the role of horizontal transmission, and cultural-evolutionary biases.

## How are scales selected?

We propose three types of selection pressure: (i) cultural-evolutionary biases are based on how many people, or which people in one's group are using the scale [161]; (ii) cognitive biases depend on how pitch is perceived by humans; and (iii) production biases depend on what is easy or difficult to sing, and physical constraints on instruments.

We can think of three examples of cultural-evolutionary biases that may apply to music: conformity, novelty and presitge biases. A bias towards conformity results in the most populous scale type being increasingly successful. In general, humans tend to synchronize [34, 162] when playing music (same meter, tonality), and over a longer time period, there is evidence of shifts towards the use of 12-TET [126, 163]. A conformity bias could arise due to other reasons, but we can probably rule out the difficulty of learning scales—studies on novel scale systems have shown that people rapidly learn their statistics [164, 165]. Novelty biases could be can describe choosing novel microtonal tunings as a means of expression [140, 151]. An example of a prestige bias is where tunings are copied from players/instruments that are acknowledged as good [46, 87].

Theories of scale selection are commonly based on cognitive biases, of which we will give a brief summary: liking harmonicity [166, 167], disliking sensory dissonance [168, 169] (or liking sensory dissonance [114]), tonal fusion [30], neurodynamic theory [170], transmissibility

[12, 132, 160, 171, 172], memory [173], and theories based on mathematical properties of scales [174–179].

Production constraints can apply to both singing and instruments [180]. When singing, large intervals cost more energy to produce than smaller ones, while small intervals are difficult to produce reliably due to limits on motor control [181]. Instruments are less constrained in this way, but still have physical limits to the number of notes, and interval range. On the other hand, instruments are constrained in how reliably they can be re-tuned to the same scale. There is a long history [152, 182] behind the practice of tuning using harmonic intervals [50, 52, 59, 67, 88], and reports exist of tuning according to the step sizes [38, 49, 51], tuning instruments visually [87, 88, 96, 183], and copying a reference instrument [46, 87].

## How can we study the evolution of scales?

Tracing the evolutionary history of scales is challenging. Scales change at rates that depend on the instruments and technology, and new scales can be invented from scratch. The evolution is potentially driven by numerous selection pressures that vary in strength across societies. Nonetheless, we propose some approaches that seem feasible.

We can try to track evolutionary trajectories of scales using historical data. By restricting the scope to a single society, one can make simplifying assumptions by using ethnographic accounts to inform models. For example, about Gamelan orchestras we know that they are typically tuned in reference to another orchestra [140], and instrument intonation changes at a steady rate (due to similar materials and environment). This can be described as an evolving network of Gamelan orchestras with edges between orchestras that influence each other. Gamelan tunings are also extensively documented in the literature, and additional tunings can be inferred from recordings.

Another approach to infer the evolutionary trajectories or selection pressures is by analyzing a population of scales. This approach requires appropriate mathematical models where multiple selection pressures can be considered in tandem. Ultimately, many models may have convergent predictions, which means that additional experiments are needed to distinguish the relative importance of different selection pressures. One limitation of this approach is the dependence on the sample of scales studied, and it is hard to construct *a priori* a representative sample of scales. In this study we control for society, but other criteria are possible. Does it matter if societies have different population sizes? How does one deal with differences in within-society variation in scales? Should we take into account frequency of scale use within a society [100, 184]? These questions will surely unravel as more data becomes available for hypothesis testing.

## Possible bias in the scale database

Relying on data collected by a limited number of ethnomusicologists, the database at this stage has sparse geographic coverage (Fig 2). Some have suggested that ethnomusicologists have a bias towards reporting findings that are considered 'interesting', thus exaggerating diversity [20]. Certain musical traditions, such as Gamelan and Thai, were very popular research topics, so they are over-represented. In contrast, there are very few quantitative measurements of scales or instruments with fewer than 5 notes, despite these being reported in many sources. Additionally, vocal scales (taken from recordings of singing) are scarce in the database, yet the voice is probably the most important instrument from an evolutionary point of view [12].

Unfortunately, in some rare instances there seem to be statistical irregularities in the reporting of tunings [46]: note that Jaap Kunst [185] reported gamelan tunings (not included in this database) where all the higher notes were exactly an octave above the lower ones, and that this

is extremely unlikely; in a study of prehistoric bone flutes where the tunings are given to an accuracy of 1 cent, one flute is recorded as having a series of equal tempered intervals: [200, 200, 200, 300] [63]. Ultimately, these biases do not void the conclusions, but the results shown here will certainly need to be updated as more data becomes available.

## Limitations to studying scale evolution

We may be witnessing a decline in diversity of scales due to the converging forces of globalization and technological change [186]. There is already evidence of homogenization, via the adoption of 12-TET [126, 163]. Thus, to understand how scales evolve, we must look to the past, but therein lies a different problem: the older the instrument, the less certain we are about how they were played. For prehistoric artefacts, we cannot be sure whether they (or their reconstructions) faithfully resemble the instrument in its original condition. So far, the only instruments that remain sufficiently intact to play are aerophones [48, 57, 63], which can produce many different scales depending on how they are played [183, 187–189]. Thus, we believe that the best source of scales is in ethnographic recordings spanning the past century [103]. It is therefore imperative that methods be developed that can faithfully infer scales from large samples of songs. Algorithms must be developed to handle low-quality recordings [190], background noise, instrument / singing segmentation [191], polyphonic stream segmentation [192], note segmentation [193], and tonal drift [194].

## Conclusion

Scales are a cornerstone of music across the world, upon which endless combinations of melodies can be generated. Surprisingly, despite a wealth of ethnomusicological research on the subject, we lacked a comprehensive, diverse synthesis of scales of the world. Here we remedy this issue, with a focus on quantitative data that will enable detailed statistical analyses about how scales evolve. Our own preliminary analyses have lent quantitative and qualitative support for the widespread (but not necessarily universal) use of the octave in some special capacity. Despite the rich diversity of scales, when put in context of how many scales are possible, what stands out in our analysis is how remarkably similar they are across the globe. Altogether, this work presents a treatise on the evolution of scales, and proposes promising avenues for future research.

## Supporting information

**S1 File.**
(PDF)

## Acknowledgments

We thank Olga Velichkina, Frank Scherbaum, Malinda McPherson for discussions and comments on the manuscript. We thank Polina Proutskova, Elizabeth Philips, Steven Brown, Patrick Savage, Kaustuv Kanti Ganguli, John Garzoli and Neil McLachlan for discussions. We thank Marcus Pearce for suggesting the name of the database.

## Author Contributions

**Conceptualization:** John M. McBride.

**Data curation:** John M. McBride, Sam Passmore.

**Formal analysis:** John M. McBride.

**Methodology:** John M. McBride, Sam Passmore.

**Supervision:** Tsvi Tlusty.

**Visualization:** John M. McBride.

**Writing – original draft:** John M. McBride.

**Writing – review & editing:** John M. McBride, Sam Passmore, Tsvi Tlusty.

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
