## [Decision Letter · Decision Letter 0]

26 Sep 2022

PONE-D-22-20874

Convergent evolution in a large cross-cultural database of musical scales

PLOS ONE

Dear Dr. McBride,

Thank you for submitting your manuscript to PLOS ONE. Your manuscript has now been reviewed by two expert reviewers, and I myself have given it a close read as well. After careful consideration, we feel that the manuscript has merit but does not fully meet PLOS ONE’s publication criteria as it currently stands. Therefore, we invite you to submit a revised version of the manuscript that addresses the points raised during the review process.

We look forward to receiving your revised manuscript.

Kind regards,

Psyche Loui

Academic Editor

PLOS ONE

Journal Requirements:

 "This work was supported by the Institute for Basic Science, Project Code IBS-R020-D1. The funders had no role in study design, data collection and analysis, decision to publish, or preparation of the manuscript."

    "We thank Olga Velichkina, Frank Scherbaum, Malinda McPherson for discussions and comments on the manuscript. We thank Polina Proutskova, Elizabeth Philips, Steven Brown, Patrick Savage, Kaustuv Kanti Ganguli, John Garzoli and Neil McLachlan for discussions. This work was supported by the Institute for Basic Science, Project Code IBSR020-D1"

    "This work was supported by the Institute for Basic Science, Project Code IBS-R020-D1. The funders had no role in study design, data collection and analysis, decision to publish, or preparation of the manuscript."

5. Please ensure that you include a title page within your main document. You should list all authors and all affiliations as per our author instructions and clearly indicate the corresponding author.

Additional Editor Comments:

Both reviewers were impressed with the work, both theoretically and methodologically. However, both reviewers raise some valid concerns and bring up additional perceptual-cognitive and music-theoretical work that should be included. Importantly, both reviewers were not so convinced that the results adequately provide support for convergent evolution per se, a sentiment with which I agree; rather, there could be similar constraints across cultures that gave rise to the prevalence of certain intervals in musical scales. Reviewer 2 offers a great starting list of such possible constraints. I would add that the growing literature on non-octave-based, artificial tuning systems such as the Bohlen-Pierce scale, and the relative ease with which humans in different cultures learn and represent pitch patterns in these scales, should be discussed as a limitation for studying scale evolution. 

As a very minor point: It was also noted that citation 33 references a pre-print, but the cited pre-print has now been published at Attention, Perception and Psychophysics in 2022. Please update the citation.

Reviewers' comments:

Reviewer's Responses to Questions

Comments to the Author

1. Is the manuscript technically sound, and do the data support the conclusions?

Reviewer #1: No

Reviewer #2: Partly

2. Has the statistical analysis been performed appropriately and rigorously?

Reviewer #1: Yes

Reviewer #2: Yes

3. Have the authors made all data underlying the findings in their manuscript fully available?

Reviewer #1: Yes

Reviewer #2: Yes

4. Is the manuscript presented in an intelligible fashion and written in standard English?

Reviewer #1: Yes

Reviewer #2: Yes

5. Review Comments to the Author

Reviewer #1: This very interesting submission presents a database of musical scales, drawn from a relatively wide range of sources representing different societies and geographical regions. This database is analyzed statistically to investigate what kinds of regularities may be found across the scales, as well as the set’s range and nature of variability. The manuscript presents some possibilities which may have given rise to both regularities and variation, across time and across geography.

The new scale database is, in this reviewer’s opinion, a fine contribution to the research literature. It appears to be larger and broader in scope than any such extant resource. Making this database, and the computational tools used to analyze it, publicly available is, to this reviewer, the greatest benefit of this study.

The analyses of the database are really quite interesting and, as the authors note, require innovative thinking as there may be no generally accepted methods for how to investigate the questions undergirding this study.

The first proposed method, comparing the database’s scales to a lognormal distribution, makes sense in some ways, but the hypotheses that scale degrees (to use the musical term) might be chosen independently from such a distribution and that that “small intervals should be uncommon” do not accord with what we know of melodic structures in general, possibly motivated by something like Fitts’ law (cf. Brinkman & Huron 2021, Ammirate & Russo 2015, Huron 2001). The notion that scales serve as organizing schemas for melodies (Dowling 1978) which are, culturally and evolutionarily, vocal in origin, would work against notions of scale degrees as independent from one another (as the authors note in a more limited way); studies of melodic entropy and other such related measures give rise to the possibility that scales arise, in part, from physiological affordances and constraints (motor, perceptual, and neurological). The second and third methods, shuffling and resampling, likewise start from assumptions which are plausible if the idealized musician involved is constructing scales as abstract entities out-of-time (perhaps as a precursor to musical composition), a view which does not seem plausible as an evolutionary mechanism. It is good to see that the authors state that “Ultimately, it is hard to say exactly how scales are chosen….”

Nonetheless, the results support what would be expected, from a multitude music-theoretic, perceptual-cognitive, and enthnomusicological crosscultural studies: “whole steps” (in the region of 200 cents), “perfect fifths” (+/- 700 cents), and octaves (+/- 1200 cents) are prounounced and notably different from what is predicted by each model. Figures 3-4 require some puzzling-through to interpret; the explanatory text is sufficiently detailed but somewhat dense.

The ”Statistics of octave scales” section was very interesting, convincing, and provocative. In the “Variation across societies…” and “Statistical analysis shows…” sections, there is much of interest, such as the clustering of equiheptatonic scales in the center of the 2D solution provided, and the quasi-circular distribution of other scales (sliced differently in the diagrams), reminiscent of spacing in 2D renderings of many phenomena (affect, musical timbre, linguistic vowel space). The lack of any interpretation of the axes of the tSNE graphs renders them opaque to much interrogation (for example, why are the Western ‘modes’ arranged as they are, given that they are rotations of the ‘same’ underlying structure?) The claim that most scales tend to be close to equidistant is tantalizing but more is needed in the way of explanation or argumentation to make the case convincingly. It’s possible that simply labelling the Y axes of the graphs in Fig. 7 would be a help in this regard.

The Discussion section begins with consideration of how diverse scales are across cultures. This section builds on two ‘case studies’, the first published (but, one expects, not widely known) on Georgian singing, and the second newly-conducted and in the supplemental information. Together, these present interesting ideas but are need expansion to sufficiently make the case; the Georgian study is itself some dozens of pages in its exploration, and the problems of dealing with pitch extraction from percussion (marimba and carillon) are nontrivial. The overall conclusion here, that “shared biases” underlie “convergent evolution”, does not seem adequately supported. One might consider neural bases (Large’s work would be a starting point, e.g. Large et al 2016) as well as cross cultural/cognitive/perceptual ones (add Dowling 1978 as a starting point) in addition to the vocal/motor approaches already mentioned in this review. “How do scales evolve over time” provides good possibilities, but is perforce speculative, as is also the case with “How are scales selected.” Many citations are given here, and the portmanteau character of the section is stimulating but could, in this reviewer’s opinion, be either greatly trimmed (reducing speculation) or expanded (to support the argument—it would take quite a bit of text to really support what’s implied here from the citation of the Partch book).

On balance, this reviewer recommends reformulating the MS to be primarily a report of a new, substantial database, and accompanying analyses, which adds to our current and widely accepted knowledge of musical scales in good ways. substantially trimming the Discussion section to reduce the speculative aspects. This is worthwhile and should be supported by publication.

However, the study’s overt goal—to set this forth as evidence for convergent _evolution_--is not supported adequately. Readers with more musical background will differ on various points as to how pitch, interval, scale, and tonality are conceptualized, defined, and utilized; readers looking for causal mechanisms—one of the key selling points for any evolutionary theory—will find references to intriguing proposals but not a convincing causal narrative. Emphasizing the rather clearer contributions of the database analysis will make for a more integrated approach, probably reduce the extremely large number of highly divergent citations to a more inter-coherent set, and allow specialists in musicology and music perception/cognition to work from the findings presented herein.

References cited

Ammirante, P., & Russo, F. A. (2015). Low-skip bias: The distribution of skips across the pitch ranges of vocal and instrumental melodies is vocally constrained. Music Perception: An Interdisciplinary Journal, 32(4), 355-363.

Brinkman, A., & Huron, D. (2021, July). Cross-Cultural Corpus Creation and Statistical Tendencies in Music. In 8th International Conference on Digital Libraries for Musicology (pp. 14-22).

Dowling, W. J. (1978). Scale and contour: Two components of a theory of memory for melodies. Psychological review, 85(4), 341.

Huron, D. (2001). Tone and voice: A derivation of the rules of voice-leading from perceptual principles. Music Perception, 19(1), 1-64.

Large, E. W., Kim, J. C., Flaig, N. K., Bharucha, J. J., & Krumhansl, C. L. (2016). A neurodynamic account of musical tonality. Music Perception: An Interdisciplinary Journal, 33(3), 319-331.

Reviewer #2: Convergent evolution in a large cross-cultural database of musical scale

This paper is clearly written, and easy to follow. It describes an analysis based on a newly assembled corpus of different types of scale data. By analyzing this data, the authors found convergence between the different types of intervals used across cultures, and also showed that scales vary as much within cultures (broadly defined) as between.

I have several broader comments, and some minor comments.

1. Section on ‘Qualitative evidence for octave equivalence’. Most of the results described in this section are evidence for the importance of (or, more precisely) the presence of octave relationships in musical scales. Evidence of the octave’s presence on musical instruments does not show that listeners perceive those notes as ‘equivalent’. Likewise, performing melodies in parallel octaves does not prove octave equivalence; many harmonies use, say, thirds in parallel, but this should not lead to the conclusion that notes separated by thirds are perceived as equivalent. I think this section could be easily re-worked to suggest that octaves are ‘important’ and/or ‘present’, but not that there is ‘octave equivalence’. Stronger evidence for octave equivalence would be that the scale system is cyclic, centered around octave transpositions, but it is not clear to me that that result falls neatly out of any of the models presented in this paper (this cyclic nature of scales was, as far as I can tell, assumed in the ‘statistics of octave scales’ section). Either way, I think the claim of octave equivalence is not necessary for the observation that octaves are a frequent interval to still be interesting.

a. The fact that the octave is equivalent is an assumption you are making to examine the statistics of ‘octave scale’. It should be stated as an assumption, rather than as a fact.

b. It could be possible to look in one octave, find what intervals are common, and see how well that predicts intervals you find in other octaves. This might provide stronger evidence for octave equivalence.

2. Page 8 – ‘humans make mistakes in discriminating intervals below ~100 cents’. There is some evidence that this threshold is precisely because (in Western listeners, the majority of those tested) sub-semitone intervals are not musically relevant. It is misleading to state this as somehow causal to how scales have developed around the world.

a. Zarate, J. M., Ritson, C. R., & Poeppel, D. (2012). Pitch-interval discrimination and musical expertise: Is the semitone a perceptual boundary?. The Journal of the Acoustical Society of America, 132(2), 984-993.

3. The high frequency of perfect fourths and perfect fifths in scales could, like octaves, be explained by tonal fusion. I think this could be mentioned earlier. Most of the introduction focuses on why the octave might be common, but most of the arguments could apply just as easily to perfect fourths and fifths, which are found in low parts of the harmonic series. This was only touched on very briefly in the discussion (page 14).

4. The relationship between melodic scales and harmony was not discussed. Conceivably harmony is a constraint on scale systems as well?

5. I did not understand the description of the database. The figure shows 931 total scales, 434 theory scales, and 497 measured scales. However, in the text a few different numbers were references (382 theory scales, 434 measured scales, etc.). Therefore, the correspondence between the figure and the text was not clear, and the relationship between the different numbers referenced in the text was likewise unclear. Potentially making a flow chart summarizing the information in the supplementary materials would avoid confusion.

6. I think the title does not match the conclusion. As the authors themselves state in the discussion, scales can be invented without referencing existing scales, and ‘evolution’ is difficult to trace with scales. I think something like ‘Convergence in a large cross-cultural database of musical scale’ would be more appropriate.

7. I think it is incorrect to say that scale evolution is relatively unconstrained (page 15). The convergence of scales around the world does give circumstantial evidence that there are constraints! The physicality of musical instruments is a constraint, for example. Ease of sung production is a constraint.

Minor comments:

8. Page 1: ‘Pitch (frequency)’ – I’d flip this and say frequency (pitch) to emphasize the measurable item (which you use in your models) rather than the percept. Why do you have ‘duration (time)’? Are you distinguishing between duration and time?

9. Page 3: ‘but is always within 10 cents’. Is there a citation for this assertion? Similar question for the statement ‘5 cents for tuning forks’ on page 8.

10. Page 2: ‘But since we perceive pitch logarithmically’ – Jacoby et al could be a useful second citation here.

11. Figure 2: Is ‘Western’ referring to South America? Or was South America included in ‘Latin America’? Either way that labeling was confusing. Maybe re-ordering the bars so they are lined up (approximately) with the appropriate area of the globe would be helpful?

12. Page 9: It was not clear initially that the first section of the result only examined measured scales. This should be stated at the beginning of the results section, not in the subsequent ‘Statistics of Octave Scales’ section.

a. Would the results of the initial section be the same if you applied the analysis to theory scales? It could be a good sanity check.

13. Figure 6: I liked this analysis and plotting!

6. PLOS authors have the option to publish the peer review history of their article (what does this mean?). If published, this will include your full peer review and any attached files.

Do you want your identity to be public for this peer review? For information about this choice, including consent withdrawal, please see our Privacy Policy.

Reviewer #1: No

Reviewer #2: No

---

## [Author Response · Author response to Decision Letter 0]

19 Feb 2023

Editor Comments:

Both reviewers were impressed with the work, both theoretically and methodologically. However, both reviewers raise some valid concerns and bring up additional perceptual-cognitive and music-theoretical work that should be included. Importantly, both reviewers were not so convinced that the results adequately provide support for convergent evolution per se, a sentiment with which I agree; rather, there could be similar constraints across cultures that gave rise to the prevalence of certain intervals in musical scales. Reviewer 2 offers a great starting list of such possible constraints. I would add that the growing literature on non-octave-based, artificial tuning systems such as the Bohlen-Pierce scale, and the relative ease with which humans in different cultures learn and represent pitch patterns in these scales, should be discussed as a limitation for studying scale evolution.

We appreciate the detailed reviews by the Editor and the two Reviewers, and give below a short summary of the major changes. We note that there seems to be some difference in interpretation of “convergent evolution”, from both Reviewers and the editor. To clear up just one point initially, the intended message of the paper was that first there is evidence of convergent evolution, which strongly implies that there are shared constraints / biases across cultures. The section, “How are scales selected?”, was an attempt at discussing these constraints / biases, but admittedly it was not very good writing, so we’ve re-written it. Other issues have been addressed in a point-by-point response further below.

 • Elaborated on what we mean by convergent evolution

 ◦ Final paragraph of the introduction

 ◦ We added a new figure (SI Fig. 5) to show that step interval size distributions have the same range in different geographical regions.

 ◦ We added a new figure (SI Fig. 12) to show that: 7-note scales from different regions all have overlapping clusters of scales; scale distance distributions are same for both within-region distances, and between-region distances.

 ◦ In “How diverse are scales?”, we now elaborate on cultural evolution, to make it clear that scale convergence can be described by two processes: cultural diffusion via trade, conquest, etc.; or evolution (change and selection). We clarify the evidence in support of our conclusion that scales have evolved convergently: out of all possible scales, humans use a small fraction of them; these scales can be characterised by their step sizes (100-400 cents), and by being close to equidistant; it is now shown in SI Fig. 5 and 12 that these two features are found in all geographical regions.

 ◦ In “How do scales evolve over time?”, we have clarified what is meant by scale evolution, by discussing mechanisms and analogies with biological evolution. Furthermore, we propose concrete next steps for creating a detailed mechanistic model of scale evolution.

 • Major revisions to “How are scales selected?”

 ◦ Reduced speculation

 ◦ Reorganized and reduced the amount of detail in the text (while keeping the references)

 ◦ Included and discussed suggested ideas / references by Editor (Loui et al., Rohrmeier) and Reviewer 1 (Dowling, Large et al.).

As a very minor point: It was also noted that citation 33 references a pre-print, but the cited pre-print has now been published at Attention, Perception and Psychophysics in 2022. Please update the citation.

We have updated the citation as requested.

Reviewer #1: This very interesting submission presents a database of musical scales, drawn from a relatively wide range of sources representing different societies and geographical regions. This database is analyzed statistically to investigate what kinds of regularities may be found across the scales, as well as the set’s range and nature of variability. The manuscript presents some possibilities which may have given rise to both regularities and variation, across time and across geography.

The new scale database is, in this reviewer’s opinion, a fine contribution to the research literature. It appears to be larger and broader in scope than any such extant resource. Making this database, and the computational tools used to analyze it, publicly available is, to this reviewer, the greatest benefit of this study.

We thank the Reviewer for their positive comments, and for taking the time to critique our manuscript.

The analyses of the database are really quite interesting and, as the authors note, require innovative thinking as there may be no generally accepted methods for how to investigate the questions undergirding this study.

The first proposed method, comparing the database’s scales to a lognormal distribution, makes sense in some ways, but the hypotheses that scale degrees (to use the musical term) might be chosen independently from such a distribution and that that “small intervals should be uncommon” do not accord with what we know of melodic structures in general, possibly motivated by something like Fitts’ law (cf. Brinkman & Huron 2021, Ammirate & Russo 2015, Huron 2001). The notion that scales serve as organizing schemas for melodies (Dowling 1978) which are, culturally and evolutionarily, vocal in origin, would work against notions of scale degrees as independent from one another (as the authors note in a more limited way); studies of melodic entropy and other such related measures give rise to the possibility that scales arise, in part, from physiological affordances and constraints (motor, perceptual, and neurological). The second and third methods, shuffling and resampling, likewise start from assumptions which are plausible if the idealized musician involved is constructing scales as abstract entities out-of-time (perhaps as a precursor to musical composition), a view which does not seem plausible as an evolutionary mechanism. It is good to see that the authors state that “Ultimately, it is hard to say exactly how scales are chosen….”

We thank the Reviewer for these insightful comments, and respond to them individually:

We agree with the Reviewer that when one considers melodic structures, small intervals are used more frequently than large ones (i.e., melodies predominantly involve scalar motion). What we mean in the manuscript is that scales with intervals below about 50 cents should be improbable, since it becomes more difficult to discriminate notes the close they are. We have updated the text to make it clear we are talking in this specific instance about “notes very close to unison (S < 50 cents)”. The suggested studies about patterns in melodic sequences are related, and good papers, but we have assessed them as ultimately out of scope.

We agree with the Reviewer that scale degrees are likely not independent from each other. We start our analysis with the log-normal distribution, not because we believe this assumption (that scale notes are i.i.d.), but because it allows us to ask what happens if we make this assumption. Our results (Fig 3) clearly show the difference between real scales and scales with notes drawn from a lognormal distribution – real scales tend to have gaps between adjacent notes of about 100-400 cents, which you do not get if you assume that the notes from a scale are generated from a log-normal distribution. In a similar way, the second and third methods show what intervals are significant given some other assumptions. We are careful not to say that these assumptions are correct, since our intention is only to show how these assumptions lead to particular results. But these three models are the simplest (most assumption-free), and most appropriate that we could think of. We are glad that this message seems to have been communicated to the Reviewer in the paragraph starting, “Ultimately, it is hard to say how scales are chosen...”.

We take the Reviewer’s view on “physiological affordances and constraints” into consideration in our major revision of the paragraph “How are scales selected?”.

Nonetheless, the results support what would be expected, from a multitude music-theoretic, perceptual-cognitive, and enthnomusicological crosscultural studies: “whole steps” (in the region of 200 cents), “perfect fifths” (+/- 700 cents), and octaves (+/- 1200 cents) are prounounced and notably different from what is predicted by each model. Figures 3-4 require some puzzling-through to interpret; the explanatory text is sufficiently detailed but somewhat dense.

To aid clarity in reading Fig. 3, we added the meanings of the marker (‘Infrequent’, ‘Frequent’) to the caption. In the text when we discuss the “Shuffle” and “Resample” results, we make it clear that the resampled distributions are shown in Fig. 3A.

The ”Statistics of octave scales” section was very interesting, convincing, and provocative. In the “Variation across societies…” and “Statistical analysis shows…” sections, there is much of interest, such as the clustering of equiheptatonic scales in the center of the 2D solution provided, and the quasi-circular distribution of other scales (sliced differently in the diagrams), reminiscent of spacing in 2D renderings of many phenomena (affect, musical timbre, linguistic vowel space). The lack of any interpretation of the axes of the tSNE graphs renders them opaque to much interrogation (for example, why are the Western ‘modes’ arranged as they are, given that they are rotations of the ‘same’ underlying structure?) The claim that most scales tend to be close to equidistant is tantalizing but more is needed in the way of explanation or argumentation to make the case convincingly. It’s possible that simply labelling the Y axes of the graphs in Fig. 7 would be a help in this regard.

We think we understand where the Reviewer is coming from. We appreciate that the Reviewer’s comments on interpretability can be anticipated since we did not discuss the t-distribution stochastic neighbor embedding (tSNE) method much at all. In the interests of making the work more accessible to its intended interdisciplinary audience, we have now discussed in the manuscript the tSNE method, and how to interpret the figures.

The tSNE method is different from dimensional-reduction methods like PCA, where the reduced dimensions can be easily interpretable. In tSNE, we calculate the distances between every pair of scales, and then a 2-dimensional embedding is found that results in similar scales being close to each other. The axes themselves are not particularly meaningful; instead, we draw meaning from distances. Since equidistant scales are found in the middle of the graph, they have the shortest distance, on average, to all other scales. The fact that Western diatonic modes are found in a circle around the equidistant scales indicates that they are all similarly distant from an equidistant scale (as can be calculated easily); and you can notice that the diatonic scales that are neighbours in this circle are most close to each other (Ionian is close to Lydian and far from Phrygian, etc.). 

The Discussion section begins with consideration of how diverse scales are across cultures. This section builds on two ‘case studies’, the first published (but, one expects, not widely known) on Georgian singing, and the second newly-conducted and in the supplemental information. Together, these present interesting ideas but are need expansion to sufficiently make the case; the Georgian study is itself some dozens of pages in its exploration, and the problems of dealing with pitch extraction from percussion (marimba and carillon) are nontrivial. The overall conclusion here, that “shared biases” underlie “convergent evolution”, does not seem adequately supported. One might consider neural bases (Large’s work would be a starting point, e.g. Large et al 2016) as well as cross cultural/cognitive/perceptual ones (add Dowling 1978 as a starting point) in addition to the vocal/motor approaches already mentioned in this review. “How do scales evolve over time” provides good possibilities, but is perforce speculative, as is also the case with “How are scales selected.” Many citations are given here, and the portmanteau character of the section is stimulating but could, in this reviewer’s opinion, be either greatly trimmed (reducing speculation) or expanded (to support the argument—it would take quite a bit of text to really support what’s implied here from the citation of the Partch book).

We agree that “How do scales evolve over time” is at times speculative, and want this point to be evident to readers. We have amended the text so that facts and speculation are more clearly distinguished.

In light of the Reviewer’s comments, we see that the section, “How are scales selected?”, is a bit rambling, lengthy, yet does not really do the topic justice. We decided to drastically shorten the text in this section, to make a sharper, simpler set of points. I added citations for Dowling and Large in the context of perceptual / neural constraints. For simplicity, and concision, we lump together perceptual and neural biases, since they closely overlap (one may argue that neural biases are a subset of perceptual biases, but this is a game of semantics).

We suspect that it is not sufficiently clear what we mean when we discuss “convergent evolution”. Analysis of the scales reveals two clear facts: step intervals are restricted to 100-400 cents; scales are close to their equidistant counterparts (of same size). Mathematically, we show that this amounts to ~99% of possible scales never being used by any society (that we have identified), while similar scales are found throughout the world. This is what we mean by “convergent evolution” – scales in different societies have somehow ended up being more similar than different. We have added two figures (SI Fig. 5, 12) to show that each geographical region exhibits the same limits on step intervals, and that scales are close to equidistant in each region. We have also updated our text description of cultural evolution of scales, in “How diverse are scales?” and the introduction summary.

On balance, this reviewer recommends reformulating the MS to be primarily a report of a new, substantial database, and accompanying analyses, which adds to our current and widely accepted knowledge of musical scales in good ways. substantially trimming the Discussion section to reduce the speculative aspects. This is worthwhile and should be supported by publication.

We have substantially trimmed the section, “How are scales selected”, and amended text where necessary to clarify the difference between fact and speculation. We have also generally gone through the Discussion section, and made changes to this effect where possible.

However, the study’s overt goal—to set this forth as evidence for convergent _evolution_--is not supported adequately. Readers with more musical background will differ on various points as to how pitch, interval, scale, and tonality are conceptualized, defined, and utilized; readers looking for causal mechanisms—one of the key selling points for any evolutionary theory—will find references to intriguing proposals but not a convincing causal narrative. Emphasizing the rather clearer contributions of the database analysis will make for a more integrated approach, probably reduce the extremely large number of highly divergent citations to a more inter-coherent set, and allow specialists in musicology and music perception/cognition to work from the findings presented herein.

We believe that the Reviewer’s assessment about the evidence for “convergent evolution” comes from subtle misunderstanding about what we mean. We now make it more clear to the reader what we mean by convergent evolution, and reiterate that the results (restricted distribution of interval step sizes; tendency to use scales that are close to equidistant) clear show convergence. We further add SI Fig 5 and 12 to show that this convergence is apparent in different geographical regions. We elaborate in the section “How diverse are scales?”, to say that the convergence is a result of both evolution (change and selection), and diffusion. When we state that this is evidence for “convergent evolution”, we mean that there is strong evidence for convergent evolution, but we do not explicitly address the degree to which convergent evolution is more/less likely than convergence via diffusion (in reality both will play a role). It is not necessary to exactly specify the mechanism of evolution to be certain that this is an example of cultural evolution – the term evolution is only used to specify a process of change and selection.

We appreciate that the Reviewer thinks that the discussion contains ‘intriguising proposals’; we hope that our proposals may form the basis of a more convincing causal narrative through future work, and hope that we are not misrepresenting the proposed models as a result of the current study.

We agree with the Reviewer that readers will differ on semantics, which is why we were careful to explicitly state in the introduction the strict definitions we use. We also gave more detail in the section “Data curation”. We appreciate that there is currently no cross-cultural / cross-discipline consensus on semantics, and we do not attempt to force one; we only wish to be clear and rigorous.

References cited

Ammirante, P., & Russo, F. A. (2015). Low-skip bias: The distribution of skips across the pitch ranges of vocal and instrumental melodies is vocally constrained. Music Perception: An Interdisciplinary Journal, 32(4), 355-363.

Brinkman, A., & Huron, D. (2021, July). Cross-Cultural Corpus Creation and Statistical Tendencies in Music. In 8th International Conference on Digital Libraries for Musicology (pp. 14-22).

Dowling, W. J. (1978). Scale and contour: Two components of a theory of memory for melodies. Psychological review, 85(4), 341.

Huron, D. (2001). Tone and voice: A derivation of the rules of voice-leading from perceptual principles. Music Perception, 19(1), 1-64.

Large, E. W., Kim, J. C., Flaig, N. K., Bharucha, J. J., & Krumhansl, C. L. (2016). A neurodynamic account of musical tonality. Music Perception: An Interdisciplinary Journal, 33(3), 319-331.

Reviewer #2: Convergent evolution in a large cross-cultural database of musical scale

This paper is clearly written, and easy to follow. It describes an analysis based on a newly assembled corpus of different types of scale data. By analyzing this data, the authors found convergence between the different types of intervals used across cultures, and also showed that scales vary as much within cultures (broadly defined) as between.

I have several broader comments, and some minor comments.

1. Section on ‘Qualitative evidence for octave equivalence’. Most of the results described in this section are evidence for the importance of (or, more precisely) the presence of octave relationships in musical scales. Evidence of the octave’s presence on musical instruments does not show that listeners perceive those notes as ‘equivalent’. Likewise, performing melodies in parallel octaves does not prove octave equivalence; many harmonies use, say, thirds in parallel, but this should not lead to the conclusion that notes separated by thirds are perceived as equivalent. I think this section could be easily re-worked to suggest that octaves are ‘important’ and/or ‘present’, but not that there is ‘octave equivalence’. Stronger evidence for octave equivalence would be that the scale system is cyclic, centered around octave transpositions, but it is not clear to me that that result falls neatly out of any of the models presented in this paper (this cyclic nature of scales was, as far as I can tell, assumed in the ‘statistics of octave scales’ section). Either way, I think the claim of octave equivalence is not necessary for the observation that octaves are a frequent interval to still be interesting.

We agree with the Reviewer that “presence” more accurately describes our results than “importance”, and that presence by itself says nothing about “octave equivalence”. We have updated the text, as suggested, to make this clear to readers. We note that instances where scale notes an octave apart have the same name (10 sources) is indeed evidence of equivalence; we clarify that this is the only type of evidence that supports equivalence that we find. 

a. The fact that the octave is equivalent is an assumption you are making to examine the statistics of ‘octave scale’. It should be stated as an assumption, rather than as a fact.

We clarify that we assume octave equivalence at the start of, “Statistics of Octave Scales”, as suggested by the Reviewer.

b. It could be possible to look in one octave, find what intervals are common, and see how well that predicts intervals you find in other octaves. This might provide stronger evidence for octave equivalence.

We thank the Reviewer for their helpful suggestion. We agree that such a test would be more appropriate for identifying octave equivalence. We note that the degree of intonation variability expected in scales would render such a test quite difficult, as we found when we did indeed try to do something like this. We did not exhaust all possible methods, or prove that it is impossible, so this is something we may revisit in subsequent work.

2. Page 8 – ‘humans make mistakes in discriminating intervals below ~100 cents’. There is some evidence that this threshold is precisely because (in Western listeners, the majority of those tested) sub-semitone intervals are not musically relevant. It is misleading to state this as somehow causal to how scales have developed around the world.

a. Zarate, J. M., Ritson, C. R., & Poeppel, D. (2012). Pitch-interval discrimination and musical expertise: Is the semitone a perceptual boundary?. The Journal of the Acoustical Society of America, 132(2), 984-993.

We appreciate the limitations of perceptual research that has only been undertaken in WEIRD societies, and revise our statement to highlight this. 

We note that the Zarate reference does not investigate cross-cultural perception, and the idea that this threshold is due to the use of semitone in Western music is speculation by the authors. We do cite two papers that do look at interval perception (see Refs 142-143), but there are too few data to generalize.

3. The high frequency of perfect fourths and perfect fifths in scales could, like octaves, be explained by tonal fusion. I think this could be mentioned earlier. Most of the introduction focuses on why the octave might be common, but most of the arguments could apply just as easily to perfect fourths and fifths, which are found in low parts of the harmonic series. This was only touched on very briefly in the discussion (page 14).

We agree with the Reviewer – although not as much as octaves, fifths and fourths both result in tonal fusion. We had thought about discussing potential reasons for increased frequency of fourths and fifths, like we did for the octave. However, we decided against it, to limit the scope of the analysis. We expect to test in another paper the explanations of why we see so many fourths and fifths. In the interest of completeness, we now mention in the intro that tonal fusion also applies to fourths and fifths.

4. The relationship between melodic scales and harmony was not discussed. Conceivably harmony is a constraint on scale systems as well?

We are not entirely sure what the Reviewer means here. For example, we discuss the idea that tonal fusion aids syncronization – this is an example where harmony (simultaneous presentation of multiple notes, rather than sequential) may result in a bias if people have a reason to synchronize. To our understanding, harmony, by itself, is does not impose additional constraints.

5. I did not understand the description of the database. The figure shows 931 total scales, 434 theory scales, and 497 measured scales. However, in the text a few different numbers were references (382 theory scales, 434 measured scales, etc.). Therefore, the correspondence between the figure and the text was not clear, and the relationship between the different numbers referenced in the text was likewise unclear. Potentially making a flow chart summarizing the information in the supplementary materials would avoid confusion.

Thank the Reviewer for pointing out this important error. We did a thorough check of the numbers and found other instances (e.g. in “Statistically significant intervals between scales”), and we believe there are no errors now in scale counts. We also found that two separate categories add up to 434, which is confusing. We have now explicitly stated whether counts are for (raw) scales or octave scales to fix this.

We would kindly point the Reviewer to SI Fig. 1 and 2, which show a flowchart with examples.

6. I think the title does not match the conclusion. As the authors themselves state in the discussion, scales can be invented without referencing existing scales, and ‘evolution’ is difficult to trace with scales. I think something like ‘Convergence in a large cross-cultural database of musical scale’ would be more appropriate.

We appreciate the point raised by the Referee, which highlights the difficulty of interdisciplinary research and the need to write clearly for all intended audiences. When we bring up the problem of ‘invention’, we do this to highlight that the mechanism of change is potentially less constrained than in, for example, biological evolution. However, in reality, scales that are ‘invented’ are typically related to other scales in some way (since they are ultimately shaped by similar constraints), and one can think of ‘invention’ as being a large change. In biology for example, large changes arise due to horizontal transfer of genes, or endosymbiosis, and these are examples of evolution. We agree that evolution may be difficult to trace with scales (one needs a lot of good quality data), but fail to see why this is relevant to whether or not cultural evolution occurs (evolution is difficult to trace in biology also, but biologists have a 100-year head-start on this). Two facts – that scales change over time, and that they are selected in some way – are sufficient for us to discuss the process of change in scales as a cultural-evolutationary process.

To avoid misunderstandings, we elaborate on the ways in which changes to scales over time can be considered evolution, and propose a roadmap to a more detailed mechanistic evolutionary model in the section, “How do scales evolve over time?”.

7. I think it is incorrect to say that scale evolution is relatively unconstrained (page 15). The convergence of scales around the world does give circumstantial evidence that there are constraints! The physicality of musical instruments is a constraint, for example. Ease of sung production is a constraint.

We appreciate that the meaning of this was not sufficiently clear. In this instance we are discussing the mechanisms of change, which appear “relatively unconstrained” when compared to biological evolution. We have revised this section to clarify what we mean by evolution of scales. The following section, “How are scales selected?” contains details about possible constraints on what scales get selected.

Minor comments:

8. Page 1: ‘Pitch (frequency)’ – I’d flip this and say frequency (pitch) to emphasize the measurable item (which you use in your models) rather than the percept. Why do you have ‘duration (time)’? Are you distinguishing between duration and time?

We have removed the brackets, and now use ‘frequency’ instead of pitch.

9. Page 3: ‘but is always within 10 cents’. Is there a citation for this assertion? Similar question for the statement ‘5 cents for tuning forks’ on page 8.

There is unfortunately no one citation that can qualify this statement for the entire database (60 references). We know that instruments like the Stroboconn could achieve very accurate measurements (~1 cents). Sometimes sources would report their errors. Otherwise, the standard means of inferring measurement error is to see to what precision (e.g. how many significant figures) the data is reported – tuning forks would usually be given to the closest 5 cents.

10. Page 2: ‘But since we perceive pitch logarithmically’ – Jacoby et al could be a useful second citation here.

We have added the suggested reference.

11. Figure 2: Is ‘Western’ referring to South America? Or was South America included in ‘Latin America’? Either way that labeling was confusing. Maybe re-ordering the bars so they are lined up (approximately) with the appropriate area of the globe would be helpful?

We rearranged the labels as suggested.

12. Page 9: It was not clear initially that the first section of the result only examined measured scales. This should be stated at the beginning of the results section, not in the subsequent ‘Statistics of Octave Scales’ section.

We have added a bit of extra text to the section, “Statistically significant intervals between scalse”, to make this more clear. We note that the captions of figures 3 and 4 both mention specifically that only measured scales are analysed.

a. Would the results of the initial section be the same if you applied the analysis to theory scales? It could be a good sanity check.

This is a good suggestion. In fact, because the theory scales are (almost) all octave scales, the subsequent section, “Statistics of Octave Scales”, along with Fig. 5, already provides this information. One can see (Fig. 5, green line) that we get the same main features for theory scales (constrained step intervals, peaks in scales at 5ths) that we found for measured scales.

13. Figure 6: I liked this analysis and plotting!

Thanks!

---

## [Decision Letter · Decision Letter 1]

29 Jun 2023

PONE-D-22-20874R1Convergent evolution in a large cross-cultural database of musical scalesPLOS ONE

Dear Dr. McBride,

Thank you for submitting your manuscript to PLOS ONE. After careful consideration, we feel that it has merit but does not fully meet PLOS ONE’s publication criteria as it currently stands. Therefore, we invite you to submit a revised version of the manuscript that addresses the points raised during the review process.

We look forward to receiving your revised manuscript.

Kind regards,

Steve Zimmerman, PhD

Associate Editor, PLOS ONE

Journal Requirements:

**Additional Editor Comments:**

Both reviewers are happy with the manuscript and indicate it is almost ready to be accepted for publication.

There are just two minor points to address. Reviewer #1 cites a recently-published paper - please feel free to comment on this paper and it's relevance. Please also address reviewer #2's concerns regarding Figure 2.

Reviewers' comments:

Reviewer's Responses to Questions

**Comments to the Author**

1. If the authors have adequately addressed your comments raised in a previous round of review and you feel that this manuscript is now acceptable for publication, you may indicate that here to bypass the “Comments to the Author” section, enter your conflict of interest statement in the “Confidential to Editor” section, and submit your "Accept" recommendation.

Reviewer #1: All comments have been addressed

Reviewer #2: (No Response)

2. Is the manuscript technically sound, and do the data support the conclusions?

Reviewer #1: Yes

Reviewer #2: Yes

3. Has the statistical analysis been performed appropriately and rigorously? 

Reviewer #1: Yes

Reviewer #2: Yes

4. Have the authors made all data underlying the findings in their manuscript fully available?

Reviewer #1: Yes

Reviewer #2: Yes

5. Is the manuscript presented in an intelligible fashion and written in standard English?

Reviewer #1: Yes

Reviewer #2: Yes

6. Review Comments to the Author

Reviewer #1: The revision addresses many of the reviewers' comments and the remaining disagreements are no more than normal for scientific discourse.

I note that a recently published paper ( Anglada-Tort, M., Harrison, P. M., Lee, H., & Jacoby, N. (2023). Large-scale iterated singing experiments reveal oral transmission mechanisms underlying music evolution. Current Biology ) directly addresses material in this submission,focally but not limited to discussion on lines 472ff.

Reviewer #2: I appreciate the attention to detail and care that the authors took in addressing the reviewer comments, and I think the paper is much clearer in its current form.

I will add two minor points, one to clarify a point from my first review, and another on one aspect that is still unclear to me:

1. "The relationship between melodic scales and harmony was not discussed. Conceivably harmony is a constraint on scale systems as well?"

- There is a world in which people would use notes within a harmonic context but not use those same notes within melodies, but to my knowledge that doesn't appear to happen. So it does seem like aspects like fusion/harmony constrain (or maybe, better put, guide) individuals to use particular notes in melodies. The added comments about fusion maybe address this issue sufficiently; I'll leave that to your discretion. I hope this clarifies my point.

2. The bar labels are still unclear to me on Figure 2. I only see one country labeled in Latin America, and many South America. Why is the label 'Latin America'?

7. PLOS authors have the option to publish the peer review history of their article (what does this mean?). If published, this will include your full peer review and any attached files.

Reviewer #1: No

Reviewer #2: No

---

## [Author Response · Author response to Decision Letter 1]

4 Jul 2023

Reviewer #1: The revision addresses many of the reviewers' comments and the remaining disagreements are no more than normal for scientific discourse.

I note that a recently published paper ( Anglada-Tort, M., Harrison, P. M., Lee, H., & Jacoby, N. (2023). Large-scale iterated singing experiments reveal oral transmission mechanisms underlying music evolution. Current Biology ) directly addresses material in this submission,focally but not limited to discussion on lines 472ff.

Response: This comment was addressed in the previous revision. See reference [160], and the two mentions of it in the text.

Reviewer #2: I appreciate the attention to detail and care that the authors took in addressing the reviewer comments, and I think the paper is much clearer in its current form.

I will add two minor points, one to clarify a point from my first review, and another on one aspect that is still unclear to me:

1. "The relationship between melodic scales and harmony was not discussed. Conceivably harmony is a constraint on scale systems as well?"

- There is a world in which people would use notes within a harmonic context but not use those same notes within melodies, but to my knowledge that doesn't appear to happen. So it does seem like aspects like fusion/harmony constrain (or maybe, better put, guide) individuals to use particular notes in melodies. The added comments about fusion maybe address this issue sufficiently; I'll leave that to your discretion. I hope this clarifies my point.

Response: This is a good comment (first two sentences) by the reviewer, and was originally in an earlier version of the manuscript, and I had even done some research on whether there are counter examples (I have not found any yet). I removed it due to the need to shorten the paper. The section “How are scales selected?” does treat with harmony through both tonal fusion, sensory dissonance, and nuerodynamic theory.

2. The bar labels are still unclear to me on Figure 2. I only see one country labeled in Latin America, and many South America. Why is the label 'Latin America'?

Response: “Latin America” was used instead of Central America or South America, since I have few examples from either area. I grouped them together just for simplification (since there are few examples, it doesn’t substantially affect any analysis). Just to clarify, ‘Latin America’ is a cultural term rather than geographic, and describes the countries in Central and South America that speak languages derived from Latin.

---

## [Editor Report · Decision Letter 2]

31 Jul 2023

Convergent evolution in a large cross-cultural database of musical scales

PONE-D-22-20874R2

Dear Dr. McBride,

We’re pleased to inform you that your manuscript has been judged scientifically suitable for publication and will be formally accepted for publication once it meets all outstanding technical requirements.

Kind regards,

Steve Zimmerman, PhD

Associate Editor, PLOS ONE
---

## [Editor Report · Acceptance letter]

28 Apr 2023

PONE-D-22-20874R1 

Convergent evolution in a large cross-cultural database of musical scales 

Dear Dr. McBride:

I'm pleased to inform you that your manuscript has been deemed suitable for publication in PLOS ONE. Congratulations! Your manuscript is now with our production department. 

Kind regards, 

on behalf of

Dr. Psyche Loui 

Academic Editor

PLOS ONE